

# Relative timing of precipitation and ocean circulation changes in the western equatorial Atlantic over the last 45 ky

Claire Waelbroeck[1], Sylvain Pichat[2,3], Evelyn Böhm[1], Bryan C. Lougheed[1], Davide Faranda[1], Mathieu Vrac[1], Lise Missiaen[1], Natalia Vazquez Riveiros[1], Pierre Burckel[4], Jörg Lippold[5], Helge W. Arz[6], Trond Dokken[7], François Thil[1], Arnaud Dapoigny[1]

[1]LSCE/IPSL, Laboratoire CNRS-CEA-UVSQ, 91198 Gif-sur-Yvette, France
[2]Laboratoire de Géologie de Lyon (LGL-TPE), Ecole Normale Supérieure de Lyon, Université de Lyon, CNRS UMR5276, 69007 Lyon, France
[3]Climate Geochemistry Department, Max Planck Institute for Chemistry, Mainz, Germany
[4]IPGP, Université Sorbonne, 75238 Paris, France
[5]Institute of Earth Sciences, Heidelberg University, Im Neuenheimer Feld 234, 69120 Heidelberg, Germany
[6]Leibniz-Institute for Baltic Sea Research Warnemünde, Seestrasse 15, 18119 Rostock, Germany
[7]Uni Research and Bjreknes Centre for Climate Research, Nygårdsgaten 112, 5008 Bergen, Norway

*Correspondence to*: Claire Waelbroeck (claire.waelbroeck@lsce.ipsl.fr)

**Abstract.** Thanks to its optimal location on the North Brazilian margin, core MD09-3257 records both ocean circulation and atmospheric changes. The latter occur locally in the form of increased rainfall on the adjacent continent during the cold intervals recorded in Greenland ice and northern North Atlantic sediment cores (i.e. Greenland stadials). These rainfall events are recorded in MD09-3257 by peaks in ln(Ti/Ca). New sedimentary Pa/Th data indicate that mid-depth western equatorial water mass transport decreased during all the Greenland stadials of the last 45 ky. Using cross-wavelet transforms and spectrogram analysis, we assess the relative phase between the MD09-3257 sedimentary Pa/Th and ln(Ti/Ca) signals. We show that decreased water mass transport between ~1300 and 2300 m depth in the western equatorial Atlantic preceded increased rainfall over the adjacent continent by 110 to 400 y at Dansgaard-Oeschger (D-O) frequencies, and by 280 to 980 y at Heinrich-like frequencies.

We suggest that the large lead of ocean circulation changes with respect to changes in tropical South American precipitation at Heinrich-like frequencies is related to the effect of a positive feedback involving iceberg discharges in the North Atlantic. In contrast, the absence of widespread ice rafted detrital layers in North Atlantic cores during D-O stadials supports the hypothesis that such a feedback was not triggered in the case of D-O stadials, with circulation slowdowns and subsequent changes remaining more limited during D-O stadials than Heinrich stadials.



## 1 Introduction

Rapid changes in ocean circulation and climate have been observed in marine sediments and polar ice cores over the last glacial and deglacial period (e.g. Johnsen et al., (1992) ; Vidal et al., (1997)). These observations demonstrate that the ocean's current mode of circulation is not unique but can rapidly switch between dramatically different states, in conjunction
with climate changes, and highlight the non-linear character of the climate system.

Documenting the precise timing and sequence of events in proxy records is a prerequisite to understand the processes responsible for rapid climate changes and improve climate models predictive skills. However, the task is complicated by the difficulty to derive precise age models for marine sediment cores. In the best cases, when marine cores are radiocarbon dated, past surface reservoir ages do not vary too much through time, and bioturbation biases remain limited (e.g. for high
sedimentation rates), dating uncertainties mainly derive from the calibration of radiocarbon ages into calendar ages. In those cases, errors are less than 150 y for the time interval 0-11 calendar ky BP (ky before 1950, noted ka), of about 400 y for the 11-30 ka interval, and of 600 to 1100 y for 30-40 ka (Reimer et al., 2013). Minimum relative dating errors between records from different marine sediment cores, or between marine and ice cores records, thus reach 500 y at the end of the last deglaciation and increase from 500 to 1500 y, for increasing ages between 11 and 40 ka. It is thus not possible to quantify
leads or lags of less than 500 y between records from different marine cores, or between marine and ice core records.

Here we take advantage of the fact that the North Brazilian margin core MD09-2357 records both ocean circulation and atmospheric changes. On the one hand, we reconstruct ocean circulation changes based on new sedimentary Pa/Th data and on epifaunal benthic isotopic ratios. On the other hand, sediment Ti/Ca measured by X-ray fluorescence (XRF) reflects past changes in rainfall on the adjacent continent (Arz et al., 1998 ; Jaeschke et al., 2007). Because Pa/Th and Ti/Ca are recorded
in the same core, their relative phasing can be examined with virtually no relative dating uncertainty.

We first present the new sedimentary Pa/Th data and their relation to changes in mid-depth water transport in the western equatorial Atlantic over the last 45 ky. We then precisely assess the relative phasing between the changes in rainfall and in ocean circulation recorded in core MD09-3257.

## 2 Material and methods

*Cores location*

Core MD09-3257 (04°14.7'S, 36°21.2'W, 2344 m) was recovered from the North Brazilian margin during R/V Marion Dufresne cruise MD173/RETRO3 at approximately the same position as core GeoB3910-2 (04°14.7'S, 36°20.7'W, 2362 m) (Arz et al., 2001; Jaeschke et al., 2007). Improved recovery of deep-sea sediments with no or little deformation of sediment layers was achieved thanks to the systematic use of the CINEMA software (Bourillet et al., 2007; Woerther and Bourillet,



2005). This software computes the amplitude and duration of the aramid cable elastic recoil, as well as the piston displacement throughout the coring phase, accounting for the length of the cable (water depth) and total weight of the coring system.

At present, the North Brazilian margin is bathed by southward flowing upper North Atlantic Deep Water (NADW) at these
depths (Lux et al., 2001; Schott et al., 2003; Rhein et al., 2015) (Fig. 1). Southward advection of dense waters formed at higher northern latitudes is channeled through the western boundary current (Rhein et al., 2015), meaning that our sediment cores are ideally located to detect changes in the transport of northern-sourced waters above 2500 m depth.

*Chronology*

Ti/Ca records from core MD09-3257 and GeoB3910-2 exhibit marked peaks corresponding to increased terrigenous input
due to enhanced precipitation and runoff from the continent (Arz et al., 1998 ; Jaeschke et al., 2007) (Fig. 2). These precipitation events are also recorded in South American speleothems, and have been shown to correspond to North Atlantic cold stadial periods (Cheng et al., 2013). Core GeoB3910-2 radiocarbon ([14]C) age model shows that increases in sedimentary Ti/Ca are indeed synchronous with decreases in South American speleothem $\delta^{18}$O (Burckel et al., 2015). Based on the observed synchronicity, composite age models of core MD09-3257 and GeoB3910-2 have been developed by using [14]C
dating for the past 35 ky, combined with the alignment of sediment Ti/Ca increases with decreases in speleothem $\delta^{18}$O for the older portion of the cores (Vazquez Riveiros et al., in prep.), thus transferring the speleothem ages beyond 35 calendar ky BP (ka) to the marine cores. In the present study, the GeoB3910-2 age scale for the 32-50 ka interval was further adjusted by precise alignment of GeoB3910-2 XRF to the MD09-3257 XRF signal (Fig. S1), thereby producing a composite record from these two nearby cores. Given that both XRF signals are virtually identical and measured at very high resolution (sampling
step ≤ 0.5 cm), the mean relative dating uncertainty between the two cores is extremely small and is less than 105 y (Fig. S1).

Here, we use XRF ln(Ti/Ca) rather than Ti/Ca because small precipitation events are more clearly marked in ln(Ti/Ca) than in Ti/Ca. We adopt the same terminology as in (Burckel et al., 2015) and define the larger ln(Ti/Ca) peaks as precipitation events PE0 to PE5, with PE0 occurring during the Younger Dryas, and PE1 to PE5 occurring during Heinrich stadials 1 to 5
(Fig. 2). What we refer to as Heinrich stadials are strictly those stadials characterized by the occurrence of iceberg discharges in the mid- to high-latitude North Atlantic. We refer to smaller ln(Ti/Ca) peaks corresponding to D-O stadials by using the Greenland stadial numbering system, as defined by (Rasmussen et al., 2014).





*Benthic isotopes*

A composite high-resolution benthic isotopic record was generated by combining isotopic data from the upper 294 cm of
core MD09-3257 (covering the last 32 ky) with isotopic data from the interval 246-451 cm in core GeoB3910-2 for the older
part of the record (Heil, 2006; Vazquez Riveiros et al., in prep.).

The $^{13}C/^{12}C$ isotopic ratio ($\delta^{13}C$, expressed in ‰ versus VPDB) of the epifaunal benthic foraminifer *Cibicides wuellerstorfi*
has been shown to record the $\delta^{13}C$ of bottom-water dissolved inorganic carbon (DIC) with minor isotopic fractionation
(Duplessy et al., 1984; Zahn et al., 1986; Schmittner et al., 2017). A water mass' initial DIC isotopic concentration is
governed by surface productivity in its formation region (i.e., the preferential consumption of $^{12}C$ by primary productivity,
thereby increasing dissolved $\delta^{13}C$), as well as temperature dependent air-sea exchanges (Lynch-Stieglitz et al., 1995). DIC

$\delta^{13}C$ subsequently decreases as deep water ages, due to progressive remineralization at depth of relatively $^{13}C$-depleted
biogenic material. As a result, DIC $\delta^{13}C$ largely follows water mass structure and circulation in the modern ocean, and *C.
wuellerstorfi* $\delta^{13}C$ ($\delta^{13}C_{Cw}$ hereafter) has been used to trace water masses and as a proxy of bottom water ventilation
(Duplessy et al. (1988) and numerous subsequent studies). A recent study further highlighted that DIC $\delta^{13}C$ more faithfully
follows water oxygen content than phosphate content (Eide et al., 2017), lending strong support to the use of $\delta^{13}C_{Cw}$ as a

proxy for bottom water ventilation, the term ventilation here referring to the transmission of oxygen-rich, atmosphere-
equilibrated water to the ocean interior.

*New sedimentary Pa/Th data*

New sedimentary ($^{231}Pa_{xs,0}/^{230}Th_{xs,0}$) measurements (excess activity ratio at the time of deposition, Pa/Th hereafter) were
produced in core MD09-3257 in order to extend the Pa/Th record of (Burckel et al., 2015) and to cover the entire time

interval 10-43 ka (Table S1). The excess activity corresponds to the fraction of each radioisotope produced in the water
column by U decay and is transferred to the sediment by adsorption on particles sinking in the water column. $^{230}Th$ and $^{231}Pa$
excess activities are calculated from bulk sediment measurement by correcting for the contribution of the detrital and
authigenic fractions (François et al., 2004; Henderson and Anderson, 2003) using a detrital ($^{238}U/^{232}Th$) value of 0.5 ± 0.1
($2\sigma$) (Missiaen et al., in revision). These excess activities are then further corrected from radioactive decay since the time of

sediment deposition. Bulk sediment measurements were performed by isotopic dilution mass spectrometry on the LSCE
MC-ICP-MS (Neptune$^{Plus}$, Thermo Fischer), following a method derived from (Guihou et al., 2010). Error bars (2 standard
deviations) on Pa/Th measurements were computed by Monte Carlo runs (Missiaen et al., in revision), accounting for the
uncertainties in Pa, Th and U measurements, as well as those of the detrital ($^{238}U/^{232}Th$) value, spike calibrations and dating.
Sedimentary Pa/Th can be used to reconstruct changes in the renewal rate of water masses overlying the core site. This tracer

has been successfully used to reconstruct past changes in deep Atlantic circulation intensity (Burckel et al. (2015) and



references therein). $^{231}$Pa and $^{230}$Th are produced at a constant Pa/Th activity ratio of 0.093 by dissolved uranium, which is homogeneously distributed in oceans. $^{230}$Th is much more particle reactive than $^{231}$Pa, as reflected by their respective residence time in the ocean (30-40 y for $^{230}$Th, 100-200 y for $^{231}$Pa (François, 2007)). $^{230}$Th is therefore rapidly removed from the water column to the underlying sediment, while $^{231}$Pa can be advected by oceanic currents. High (low) flow rates

therefore result in high (low) $^{231}$Pa export and hence low (high) sedimentary Pa/Th ratio. In contrast to $\delta^{13}C_{Cw}$, which records the DIC $\delta^{13}C$ of bottom waters at the core site, sedimentary Pa/Th does not reflect the flow rate at the seabed but that of a water layer of a few hundreds to more than 1000 m above the seafloor (Thomas et al., 2006).

Several potential caveats of the proxy were tested. In particular, $^{231}$Pa has a higher affinity for opal than for the other types of particles (Chase et al., 2002) so that high opal fluxes can result in high sedimentary Pa/Th values even in the presence of

lateral advection. Similarly, areas of very high vertical particle flux, such as the Atlantic off Western Africa, are characterized by high Pa/Th values (Yu et al., 1996; François, 2007; Lippold et al., 2012). Recent studies have shown that the caveats that may apply to this proxy in some areas do not apply to the western tropical Atlantic region. More specifically, a study including core top material from the western tropical Atlantic margin and using a 2-D model (Luo et al., 2010) showed that the measured Pa/Th vertical profile is consistent with a dominant role of the overturning circulation, rather than

particle scavenging, thereby demonstrating that Pa/Th can be used to record changes in water mass overturning rates in that region (Lippold et al., 2011). However, because there are large increases in terrigenous material deposition on the north-east Brazilian margin during the last glacial, we carefully evaluated/assessed if increased terrigenous deposition may have impacted Pa/Th values.

The $^{230}$Th-normalized $^{232}$Th flux, hereafter simply referred to as the $^{232}$Th flux, is indicative of the vertical terrigenous flux to

the core site. Besides the main precipitation events PE0 to PE4, there is no significant correlation between the Pa/Th ratio and the $^{232}$Th flux ($r = 0.21$, $p = 0.07$) (Fig. S2 and S3). In contrast, because the correlation between Pa/Th and the $^{232}$Th flux becomes significant ($r = 0.57$, $p \ll 0.001$) when including the main precipitation events (Fig. S3), the high Pa/Th values observed during PE0 to PE4 could be partly caused by increased terrigenous flux and should be interpreted with caution (empty symbols in Fig. 2). Note that a possible terrigenous influence during the main precipitation events does not preclude

that the high Pa/Th values during these periods reflect an almost halted oceanic circulation above the core site. Indeed, Pa scavenging by boundary scavenging can be intensified in times of reduced overturning circulation due to boundary scavenging becoming the main control on sedimentary Pa/Th.

Another source of possible biases in Pa/Th results from variations in opal flux (Chase et al., 2002). However, the North Brazilian margin is known for its low siliceous primary production (Arz et al., 1998). This is confirmed by $^{230}$Th-normalized

opal flux measurements in MD09-3257, which are below 0.06 g cm$^{-2}$ ky$^{-1}$ (Fig. S3). Moreover, outside of precipitation events PE0 to PE4, there is no correlation between Pa/Th and opal flux (Fig. S3). In conclusion, we may consider that



outside of the main precipitation events, our Pa/Th record can be interpreted in terms of changes in the strength of overturning circulation above MD09-3257 coring site.

### *Cross-correlation and wavelet analysis*

Assuming that there exists a constant phase shift between two time series over their entire length, one can perform a simple
cross-correlation analysis and compute how the correlation coefficient between the two time series varies as a function of the time lag between the two series (e.g. Davis and Sampson (1986)).

We normalized (i.e. subtracted the mean and divided by the standard deviation) and resampled the time series Pa/Th, $\delta^{13}C_{Cw}$ and ln(Ti/Ca) to a common age scale using scenarios with constant time steps varying between 50 and 500 y. We then used the R function cor.test (R package *stats* version 3.2.2) for correlation between paired series (R script in supplementary
material) to compute the Spearman correlation coefficient between all pairs of the three time series, after having shifted one with respect to the other by increments of the time step.

Another approach consists of classical spectral analysis methods that examine the coherence and phase between two time series in frequency space, such as Fourier transforms. Fourier transforms involve decomposing a signal in infinite-length oscillatory functions (such as sine waves). As such, these methods also rely on the assumption that the decomposition of
each signal into characteristic frequencies is valid over its entire length, i.e. that the underlying processes are stationary in time.

In contrast, wavelet analysis can be used to decompose a time series into time-frequency *space*, rather than frequency space, that is, to determine both the dominant modes of variability and how these modes vary in time (Torrence and Compo, 1998). To do so, the wavelet transform decomposes the signal into a sum of small wave functions of finite length that are highly
localized in time. Wavelet transform can thus describe changes in frequencies along the studied time series and are particularly relevant for dealing with climatic signals, since they are in essence not stationary in time, but in constant evolution in response to external forcing (i.e. insolation changes), and as a result of internal climate variability.

Given two times series $X$ and $Y$, with wavelet transforms $W^X$ and $W^Y$, the cross-wavelet spectrum is defined as $W^{XY} = W^X W^{Y*}$, where $W^{Y*}$ is the complex conjugate of $W^Y$ (Torrence and Compo, 1998). Similarly to Fourier coherency, which is used
to identify frequency bands in which two time series are related, the wavelet coherency was developed to identify both frequency bands and time intervals over which the two time series are related. The wavelet coherence between two time series is defined as the square of the smoothed cross-wavelet spectrum normalized by the smoothed individual wavelet power spectra (Torrence and Webster, 1999). This definition resembles that of a traditional correlation coefficient, i.e. wavelet coherence ranges between 0 and 1, and may be viewed as a localized correlation coefficient in time-frequency space
(Grinsted et al., 2004).





Analogous to Fourier cross-spectral analysis, the phase difference between two time series can also be computed using a cross-wavelet spectrum. The complex argument arg($W^{XY}$) can be interpreted as the local relative phase between $X$ and $Y$ in time-frequency space (Grinsted et al., 2004).

In our present study we use the software developed by (Grinsted et al., 2004) to compute the cross-wavelet spectrum, coherence and relative phase between our time series that were normalized and resampled as described before. To test for the persistence of regions of high cross-wavelet coherence, we ran all cross-wavelet analyses 1000 times for each dataset pair (i.e. a Monte Carlo approach). For each of the 1000 runs, each time data point was randomly sampled, whereby a Gaussian distribution of each data point's value (based on the measurement uncertainty) is used to weight the random sampling. Mean and standard deviation values for the coherence and phase direction were calculated using the 1000 runs.

## 3 Results

### 3.1 Ocean circulation proxy records

The Pa/Th record of core MD09-3257 now covers the entire 10-43 ka time interval, encompassing the Younger Dryas (YD) and the last four Heinrich stadials (Fig. 2). We have increased its temporal resolution over the time interval 31-38 ka comprising Dansgaard-Oeschger (D-O) events 5 to 8, with respect to the rest of the studied period, in order to examine Atlantic circulation dynamics during D-O events.

Pa/Th data exhibit systematic increases in conjunction with stadials, even if Pa/Th data points that are potentially biased towards elevated values by increased terrigenous input (empty symbols) are discarded. Pa/Th data thus indicate that the renewal rate of the water mass overlying the site decreased during stadials. More specifically, transport of the overlying water mass decreased not only during the YD and Heinrich stadials, but also during practically all D-O stadials. Among D-O stadials, the Pa/Th increase is well marked for GS-7, GS-8 and GS-11, but the signal is too noisy to provide a clear picture for GS-6. This noisy Pa/Th signal is very likely due to sediment reworking, given that the $\delta^{13}C_{Cw}$ record is also noisy over this section of the core. Also, it is noteworthy that no precipitation event is recorded in MD09-3257 or GeoB3910 during GS-10 (Fig. 2). There is no clear decrease in the well-dated El Condor (Cheng et al., 2013) speleothem $\delta^{18}O$ records associated with GS-10 either, in contrast with the other Greenland stadials (Burckel et al., 2015). It would seem that there was no apparent increase in precipitation during GS-10 over tropical South America, in contrast to all other GS of the past 40 ky. Overall, longer stadials seem to be associated with larger increases in Pa/Th than shorter stadials.

The $\delta^{13}C_{Cw}$ composite record varies in concert with Pa/Th, with high values indicating the presence of well-ventilated waters during the Holocene and interstadials, and low values indicating a marked reduction in water ventilation during stadials at ~2350 m in the western equatorial Atlantic (Vazquez Riveiros et al., in prep.).



### 3.2 Relative timing of Pa/Th, $\delta^{13}C_{Cw}$ and Ti/Ca

Pa/Th, $\delta^{13}C_{Cw}$ and Ti/Ca are recorded in the same core or in two cores from the same location, which could be precisely aligned through high resolution XRF signals. This situation provides ideal conditions to examine the relative phasing of one proxy with respect to another. Pa/Th and Ti/Ca are recorded in the same core, so their relative phasing can consequently be

examined with the smallest possible relative dating uncertainty, whereby the only remaining source of uncertainty is bioturbation. The situation is practically the same when examining $\delta^{13}C_{Cw}$ versus Ti/Ca or $\delta^{13}C_{Cw}$ versus Pa/Th. Apart from the unavoidable uncertainty introduced by bioturbation, the relative dating uncertainty between the $\delta^{13}C_{Cw}$ composite record and any MD09-3257 record is null over 0-32 ka, and amounts to 102 y on average over the 32-50 ka time interval (Fig. S1). In what follows, we assess the relative phasing between Pa/Th, $\delta^{13}C_{Cw}$ and Ti/Ca, using all Pa/Th data points (including

Pa/Th values susceptible to be partially impacted by large particle fluxes) in order to have sufficient data to examine periodicities ranging from 1000 to 6000 y. In doing so, we assume that changes in particle fluxes may affect the amplitude of the Pa/Th changes, rather than the timing of these changes. In the following text, we show that excluding the Pa/Th values susceptible to be partially impacted by large particle fluxes does not change our conclusions concerning D-O periodicities (i.e. 1000 to 3000 y).

### 3.2.1 Average relative phases

We first apply the simple stationary cross-correlation approach to examine how the correlation coefficient of Pa/Th versus ln(Ti/Ca), of $\delta^{13}C_{Cw}$ versus ln(Ti/Ca), and of $\delta^{13}C_{Cw}$ versus Pa/Th, varies as a function of the lag between the different time series (Fig. 3). Prior to computing the correlation coefficient, the three time series were resampled with a time step of 100 y and normalized.

Taken at face value, these results indicate that Pa/Th leads ln(Ti/Ca) (or Ti/Ca) by $200 \pm 100$ y, that there is no significant phase shift between $\delta^{13}C_{Cw}$ and Ti/Ca, and that $\delta^{13}C_{Cw}$ lags Pa/Th by $200 \pm 100$ y (Table S2). The uncertainty of $\pm 100$ y directly results from the adopted sampling step of 100 y. In addition, in order to assess the robustness of these results, we applied the same approach to the upper half and lower half of the records. In all cases, we obtained $\delta^{13}C_{Cw}$ lags over Pa/Th of 200 y, and Pa/Th leads over ln(Ti/Ca) of 200 or 300 y, while the phase shift between $\delta^{13}C_{Cw}$ and Ti/Ca remained between -

100 and +100 y.

Although this simple method has been applied to climatic time series in previous studies (Langehaug et al., 2016; Henry et al., 2016), such results must be interpreted with caution, since the method has been designed for signals that are stationary in time and is, therefore, not suitable for climatic signals.



### 3.2.2 Wavelet transforms

The non-stationary character of climatic signals over the last 40 to 45 ky is particularly pronounced. Different typical pseudo-periodicities can be identified for Heinrich and D-O stadials. In the case of Heinrich stadials (corresponding to our main precipitation events), the interval 11.7-49 ka comprises 5 pseudo-cycles that are ~6 to 9 ky long (Fig. 2), such that

Heinrich stadials over the studied interval are characterized by an average pseudo-periodicity of about 7 ky. Concerning D-O events, the interval located between HS3 and HS4 (32.5-38.1 ka) comprises 3 pseudo-cycles that are ~1.2, 1.5 and 3 ky long (Fig. 2), yielding an average pseudo-periodicity of about 1.8 ky.

We computed the cross-wavelet spectrum, coherence and phase between ln(Ti/Ca) and Pa/Th (Fig. 4), between $\delta^{13}C_{Cw}$ and Pa/Th (Fig. 5), and between $\delta^{13}C_{Cw}$ and ln(Ti/Ca) (Fig. 6), using the software of (Grinsted et al., 2004). The 95% confidence

level against red noise is shown as a thick contour line. Relative phases are plotted only for coherences higher than 0.5 (< 0.5 is masked as dark blue). Note that the shaded areas in Fig. 4-6 correspond to the region of the wavelet transform graphs where the edge effects due to the finite length of the time series limit the ability to carry out cross-wavelet analysis. These regions are not considered in our interpretations.

To assess the robustness of our results, we repeated the cross-wavelet transform for different interpolation resolutions

ranging from 50 to 500 y and could verify that the features corresponding to the 95% confidence level against red noise for a time step of 100 y are still present at roughly the same time and frequency for other time steps (e.g. see Fig. S4 for results obtained for a time step of 400 y).

Moreover, we ran a spectrogram analysis in order to confirm our wavelet results and avoid any over-interpretation (see Fig. S5 and explicative caption). Unlike the wavelet, the spectrogram analysis is based on a finite time Fourier transform that

spans different periods. It provides therefore an alternative base to check wavelet-based results. These tests confirmed the wavelet results for periods comprised between 1 and 6 ky. Beyond 6 ky, wavelet results could not be confirmed by spectrograms due to the short duration of the analyzed records. We do thus not discuss periodicities longer than 6 ky in what follows.

With this in mind, the following regions of significant mean coherence and well-defined mean relative phases can be

identified in the cross-wavelet graphs between Pa/Th and ln(Ti/Ca) produced by 1000 Monte Carlo runs (Fig. 4, middle panels): a coherence higher than 0.5 is found for periodicities around 2000 y (ranging from ~1000 to 3000 y) over the time interval ~28-40 ka, and for periodicities around 5000 y (~4000 to 6000 y) over ~25-40 ka. Computing the average phases over each of these two regions, we find that Pa/Th leads ln(Ti/Ca) by 259 ± 140 y (1σ) for periodicities of 1000 to 3000 y over 28-40 ka, and by 631 ± 345 y (1σ) for periodicities of 4000 to 6000 y over 15-40 ka (Table 1).

The cross-wavelet graph between $\delta^{13}C_{Cw}$ and Pa/Th displays slightly different regions of high mean coherence (Fig. 5, middle panels). Examining the same frequency bands as for Pa/Th versus ln(Ti/Ca), we find mean coherences higher than



0.5 for periodicities around 2000 y over ~28-40 ka, and for periodicities around 5000 y over ~15-40 ka. Average phases for these regions indicate that $\delta^{13}C_{Cw}$ lags Pa/Th by 279 ± 244 y (1σ) for periodicities of 1000 to 3000 y over 28-40 ka, but that the lag of $\delta^{13}C_{Cw}$ with respect to Pa/Th for periodicities of 4000 to 6000 y over ~15-40 ka is not significant (Table 1).

Finally, the regions characterized by mean coherences higher than 0.5 between $\delta^{13}C_{Cw}$ and ln(Ti/Ca) are similar to those

observed in the graph for Pa/Th and ln(Ti/Ca) (Fig. 6, middle panels). However, the average phases between $\delta^{13}C_{Cw}$ and ln(Ti/Ca) over these regions are not significantly different from zero (Fig. 6d and Table 1), indicating that decreases in $\delta^{13}C_{Cw}$ are in phase with increases in ln(Ti/Ca) within uncertainties.

Note that uncertainties for leads and lags given in Table 1 are computed as the propagation of two uncertainties: (i) the standard deviation of the mean relative phases over the given time-frequency region (Fig. 4-6d), and (ii) the median value of

the standard deviation computed by 1000 Monte Carlo runs over the same time-frequency region (Fig. 4-6f). In the case of relative phases between Pa/Th or ln(Ti/Ca) and $\delta^{13}C_{Cw}$, we also accounted for the additional error due to the combining of MD09-3257 and GeoB3910 $\delta^{13}C_{Cw}$ records.

We also applied the aforementioned cross-wavelet method only to the subset of Pa/Th data points not affected by large particle fluxes. For the periodicities between ~1000 and 4000 y, the results obtained using this subset (Fig. S6) are virtually

unchanged with respect to the results obtained using the entire data set, but coherence decreases for longer periodicities. This decrease is expected because the suppressed data points are all located in the main precipitation events, and hence correspond to the YD and Heinrich stadials.

## 4 Discussion

### 4.1 Reconstructed ocean circulation changes over the last 45 ky

Oceanographic studies have shown that the southward transport of northern-sourced waters in the equatorial Atlantic mainly takes place between ~1300 and 4000 m depth in a ~ 100 km wide Deep Western Boundary Current (DWBC) (Lux et al., 2001; Rhein et al., 2015). Using hydrographic, geochemical, and direct velocity measurements acquired in 1993 to inverse an ocean circulation model, (Lux et al., 2001) estimated that the volumetric flow of upper North Atlantic Deep Water (NADW) occupying water depths between ~1300 and 2300 m at 4.5° S within the DWBC is 11.2 Sv (1 Sv = $10^6$ m$^3$ s$^{-1}$).

This estimate is in good agreement with the 10.9 Sv estimated by (Schott et al., 2003) based on data from 13 shipboard current-profiling sections taken during the World Ocean Circulation Experiment period 1990-2002.

Our data show that outside of the main precipitation events, the total vertical particle flux did not vary much (remaining within 25.1 ± 3.6 g m$^{-2}$ y$^{-1}$, 1σ) (Fig. S2). The Pa/Th values of these interstadials are similar or slightly higher than that of the





Late Holocene (Fig. 2), suggesting that the transport of the water mass overlying the core MD09-3257 site was also of ~10 Sv during these interstadials.

It is more difficult to translate the observed increases in Pa/Th during stadials into quantified decreases in water mass transport. However, our new data bring additional observational constraints on the Atlantic circulation changes associated

with last glacial millennial climate changes. Two recent studies have indicated that decreases in northern-sourced deep water flow took place during each stadial. On the one hand, increases in Pa/Th during each stadial of the last glacial have been observed at a very deep western North Atlantic site located at ~42 °N and 4500 m depth (Henry et al., 2016). On the other hand, reconstructions of water corrosiveness in a South Atlantic core located at ~44°S and 3800 m depth indicate the absence of northern-sourced deep water at that site during stadials, whereas nearly all interstadials of the last 60 ky are characterized

by incursions of northern-sourced deep water into the deep South Atlantic (Gottschalk et al., 2015). Together with these independent results, our results indicate that decreases in both the flow rate and extension of northern-sourced deep waters during stadials were not limited to very dense waters circulating at 3800 m or deeper, but also affected water mass transport above 2350 m in the western equatorial Atlantic.

## 4.2 Relative timing of Pa/Th, $\delta^{13}C_{Cw}$ and Ti/Ca

### 4.2.1 Stationary cross-correlation versus cross-wavelet results

Cross-wavelet graphs (Fig. 4-6) show that at the site of MD09-3257, significant coherence and well-defined relative phases between $\delta^{13}C_{Cw}$, Pa/Th and Ti/Ca can be identified in only some regions of the time-frequency space. For instance, when examining the relative phase between $\delta^{13}C_{Cw}$ and Pa/Th over the interval 10-43 ka, a meaningful relative phase can be identified only over ~28-40 ka at D-O frequencies (i.e. periodicities of 1000 to 3000 y) (Fig. 5, Table 1). Furthermore, cross-

wavelet results indicate that $\delta^{13}C_{Cw}$ lags Pa/Th by 279 ± 244 y at D-O frequencies, and that decreases in $\delta^{13}C_{Cw}$ are in phase with increases in Pa/Th for periodicities of 4000 to 6000 y (i.e. closer to Heinrich periodicities). This is in contrast with the constant 200±100 lag of $\delta^{13}C_{Cw}$ with respect to Pa/Th obtained by cross-correlation between the two same time series (Fig. 3), and confirms that the latter method yields imprecise and unreliable results when applied to climatic signals.

Cross-correlation has nevertheless been recently applied to climatic signals (Langehaug et al., 2016), including Pa/Th and

$\delta^{13}C_{Cw}$ records from the last glacial (Henry et al., 2016). In the latter study, cross-correlation between marine records from two deep Bermuda Rise cores and the NGRIP ice oxygen isotopic record was used to infer that deep Bermuda Rise $\delta^{13}C_{Cw}$ led NGRIP by approximately two centuries and that Pa/Th was approximately in phase with NGRIP over the interval 25-60 ka (Henry et al., 2016). The authors further inferred that Pa/Th lags $\delta^{13}C_{Cw}$ by two centuries at their deep Bermuda Rise site. However, as shown here, cross-correlation does not yield reliable results when applied to climatic signals of the last glacial.

Moreover, the inferred relative phases between the marine and NGRIP records are much smaller than the dating error for



each individual time series, and, therefore, also the relative dating error of one time series with respect to the other. In summary, the application of stationary cross-correlation techniques and incomplete consideration of geochronological uncertainty casts doubt on the conclusions of the aforementioned studies.

### 4.2.2 Lead of Pa/Th with respect to ln(Ti/Ca)

Our cross-wavelet results show that MD09-3257 Pa/Th leads ln(Ti/Ca) by 259 ± 140 y (1σ) for periods of 1000 to 3000 y during the ~28-40 ka time interval, and by 631 ± 345 y (1σ) for periods of 4000 to 6000 y during ~15-40 ka (Table 1). Periods of 1000 to 3000 y correspond to pseudo periodicities typical of D-O stadials, while periods of 4000 to 6000 y are close to those of Heinrich stadials. It can be noted that the cross-wavelet results for D-O periodicities are significant only for the ~28-40 ka time interval, which indeed corresponds to the interval of our records for which D-O events are best recorded.

It is important to examine if the observed relative phases could be an artifact due to bioturbation. It has been shown that smaller particles are more likely to be transported by bioturbation than larger particles (Wheatcroft, 1992; McCave, 1988; DeMaster and Cochran, 1982) and that this results in fine particles having apparent younger ages than coarse particles from the same depth in a core (Brown et al., 2001; Sepulcre et al., 2017).

Sedimentary Pa/Th is measured on bulk sediment samples, with dissolved Pa and Th being more readily adsorbed on small
particles because of their higher surface to volume ratio (Chase et al., 2002). It has been shown that 50-90% of [230]Th excess inventory is found in particles smaller than 10 μm (Kretschmer et al., 2010; Scholten et al., 1994 ; Thomson et al., 1993). It is therefore reasonable to assume that the Pa/Th signal is mostly carried by small particles (< 100 μm).

Assessing the size fraction corresponding to the Ti/Ca signal is more complicated. XRF measurements show that the marked changes in ln(Ti/Ca) recorded in MD09-3257 result from sharp changes in both Ca and Ti concentration in the sediment. Ca
is a component of marine calcite and aragonite, and is thus mainly carried by large size fractions of the sediment (> 60 μm). However, previous studies have shown that changes in marine carbonate production and dissolution between 2000 and 3000 m in the western tropical Atlantic were relatively small over the last glacial (Rühlemann et al., 1996 ; Gerhardt et al., 2000). Therefore, the sharp decreases in Ca concentration during stadials result from dilution of marine carbonates by increased input of terrigenous material and the ln(Ti/Ca) signal is driven by changes in terrigenous input, rather than by changes in
marine carbonate production or dissolution. It is difficult to assess in which particle size fraction Ti is mostly concentrated. Knowing that Rb and K are typical constituents of clays and, therefore, characteristic of small grain sizes, we verified if a phase shift could be detected between the XRF Ti signal on the one hand, and the XRF Rb and K signals on the other hand. We found no relative offset between Ti and Rb and almost no relative offset between Ti and K, with the inflexion point in the K signal taking place 0.05 cm deeper than in the Ti signal. Given that core MD09-3257 sedimentation rates range from 6
to 14 cm/ky, 0.05 cm corresponds to 5 to 10 y and is thus completely negligible with respect to the observed phase shifts



between ln(Ti/Ca) and Pa/Th. We may thus consider that ln(Ti/Ca) and Pa/Th are both carried by small particles and that the observed phase shifts between these two signals are not the result of bioturbation.

Finally, if, against all likelihood, bioturbation were responsible for a lead of Pa/Th with respect to ln(Ti/Ca), such a lead would be independent from the examined periodicity. We may, therefore, reasonably assume that the observed lead of Pa/Th

with respect to ln (Ti/Ca) is not an artifact resulting from bioturbation.

We compute a $631 \pm 345$ y ($1\sigma$) lead for Pa/Th over ln(Ti/Ca) by cross-wavelet analysis for frequencies close to those characterizing Heinrich stadials. This lead is comparable to the relative phase previously estimated between MD09-3257 Pa/Th and Ti/Ca at the onset of HS4 ($690 \pm 180$ y) and HS2 ($1420 \pm 250$ y) respectively, using a completely independent approach (Burckel et al., 2015). The large lead of Pa/Th with respect to ln(Ti/Ca) is indeed clearly visible for the YD and all

Heinrich stadials, except HS1 (Fig.2). The apparent synchronicity of the Pa/Th and ln(Ti/Ca) signals at the onset of HS1 in core MD09-3257, as also recently observed in another core of the northern Brazilian margin (Mulitza et al., 2017), suggests that the sequence of events was different than at the beginning of the other Heinrich stadials. Also of note are the anomalously high sedimentary Pa/Th values at the onset of HS1, which are likely related to extremely high terrigenous particle fluxes (Fig. S2). Records of organic matter proxies from core GeoB3910 and a neighboring North Brazilian core

show that Heinrich stadials were characterized by a specific sequence of events, whereby increased rainfall firstly led to an initial outwash of organic matter-depleted mineral matter (e.g. exposed shelf sediments or eroded topsoil), after which vegetation cover subsequently developed and reduced erosional processes (Dupont et al., 2010; Jennerjahn et al., 2004). We suggest that lower sea levels during the last glacial maximum caused larger area of shelf to be exposed to erosion, resulting in a larger initial outwash for PE1 than for the other main precipitation events. As a consequence, it seems likely that

exceptionnally large vertical particle fluxes temporarily overprinted the impact of water mass transport changes on MD09-3257 sedimentary Pa/Th ratio at the onset of PE1.

Our results further indicate that a significant lead of Pa/Th with respect to ln(Ti/Ca) is also present at D-O frequencies. Moreover, the lead of Pa/Th with respect to ln(Ti/Ca) is markedly shorter at D-O frequencies ($259 \pm 140$ y) than at Heinrich frequencies ($631 \pm 345$ y).

Climate models do indeed simulate a southward shift of the low latitudes atmospheric convection zone (Intertropical Convergence Zone, ITCZ) and associated maximum in precipitation in response to an AMOC slowdown after a few hundred years (see (Kageyama et al., 2010) for a review). Furthermore, our results are consistent with the mechanism proposed by (Burckel et al., 2015) to explain the larger lead of AMOC slowdowns during Heinrich stadials than D-O stadials with respect to precipitation events over tropical South America. In this scenario, AMOC slowdowns are progressively amplified through

a positive feedback linking the decrease in deep water formation to subsurface warming at high northern latitudes (Mignot et al., 2007; Alvarez-Solas et al., 2013), leading in the case of Heinrich stadials, to erosion of ice shelves and iceberg





discharges, which in turn reinforce the initial AMOC slowdown. In contrast, AMOC slowdowns associated with D-O stadials would not trigger such a positive feedback loop and would hence remain limited.

Alternatively, or in addition to an actual lead of the changes in AMOC with respect to precipitation events over tropical South America, another factor could induce a lead of Pa/Th with respect to ln(Ti/Ca) in core MD09-3257. It has been shown

that the North Brazil Current (NBC) is able to transport terrigenous material laterally (Allison et al., 2000). Also, different studies have shown that a weakening of the AMOC is associated with a decrease of the NBC transport, taking place not only on decadal timescales (Zhang et al., 2011), but also during the YD and HS1 (Arz et al., 1999; Wilson et al., 2011). Based on this evidence, a recent study suggested that a reduced NBC during HS1 allowed the enhanced input of terrigenous material to settle down on the continental margin offshore North-East Brazil, instead of being transported northward (Zhang et al.,

2015). It thus seems possible that terrestrial input would be deviated northward as long as the NBC is vigorous and reach the core site only once the NBC and AMOC are sufficiently reduced, thereby yielding a time-delayed peak in ln(Ti/Ca). If this is the case, the lag of the terrestrial input signal with respect to the Pa/Th signal would be partially or totally caused by the impact of the NBC on terrigenous material deposition (Zhang et al., 2015). The exceptionnal synchronicity of the onset of terrigenous influx and AMOC slowdown at the beginning of HS1 could, therefore, be due to the exceptionnally large fluxes

of large grain size material eroded from the proximal exposed shelf during low eustatic sea level, which would have rained down through the water column, even before full reduction of the NBC.

However, in the absence of direct measurements of the NBC velocity and vertical particle flux on the North-East Brazilian margin, the actual delay of terrestrial input with respect to NBC slowdown remains speculative.

### 4.2.3 Lag of $\delta^{13}C_{Cw}$ with respect to Pa/Th

Our cross-wavelet results show that at MD09-3257 site, $\delta^{13}C_{Cw}$ lags Pa/Th by 279 ± 244 y (1σ) at D-O frequencies over ~28-40 ka (Table 1).

$\delta^{13}C_{Cw}$ is measured using > 150 μm foraminifers (Vazquez Riveiros et al., in prep.) and is thus carried by much larger particles than the Pa/Th signal. In that case, differential bioturbation mixing processes would lead to Pa/Th being carried by sediment material younger than the epibenthic foraminifers sampled within the same depth interval. Bioturbation may thus

induce an artificial lead of Pa/Th with respect to $\delta^{13}C_{Cw}$. Knowing that the sedimentation rate of MD09-3257 and GeoB3910 varies between 6 and 14 cm over the interval 28-40 ka, a 280 y lead translates into a downward shift of 2 to 4 cm in the sediment column, which seems a plausible effect of bioturbation.

In conclusion, the lag of $\delta^{13}C_{Cw}$ with respect to Pa/Th at D-O frequencies during ~28-40 ka is likely an artifact resulting from the differential bioturbation of fine and coarse particles. The same differential bioturbation processes likely also affect the





relative phase between $\delta^{13}C_{Cw}$ and ln (Ti/Ca). We will thus not further discuss the results of the cross-wavelet analyses involving $\delta^{13}C_{Cw}$.

## 5 Conclusions

New sedimentary Pa/Th data from core MD09-3257 located on the North Brazilian margin (~4°S, 36°W) at ~2350 m depth
indicate decreases in water mass transport above the core site during all Greenland stadials of the last 45 ky. Together with two other recent studies (Gottschalk et al., 2015 ; Henry et al., 2016), these results demonstrate that all stadials of the last 45 ky were not only characterized by decreases in flow rate and extension of northern-sourced waters below 3800 m depth, but also by decreases in mid-depth water mass transport in the western equatorial Atlantic.

Due to its exceptional location, core MD09-3257 records both ocean circulation and atmospheric changes. Ocean circulation
changes induce changes in sedimentary Pa/Th and $\delta^{13}C_{Cw}$, whereas changes in precipitation over the adjacent continent induce changes in marine sediments Ti/Ca.

Using cross-wavelet transforms and spectrogram analysis, we were able to precisely and robustly assess the relative phase between MD09-3257 sedimentary Pa/Th and ln(Ti/Ca) signals over the interval 10-43 ka with minimal uncertainty, thanks to both signals being recorded in the same sediment core. We show that Pa/Th leads ln(Ti/Ca) by 259 ± 140 y (1σ) at D-O
frequencies over 28-40 ka, and by 631 ± 345 y (1σ) for periodicities close to Heinrich periodicities (4000 to 6000 y) over 15-40 ka.

The relative lead of Pa/Th over ln(Ti/Ca) is indeed clearly visible for the YD and all Heinrich stadials, except HS1. We suggest that the apparent synchronicity of the Pa/Th and ln(Ti/Ca) signals at the onset of HS1 in core MD09-3257 may be explained by low eustatic sea level and exceptionnally large fluxes of material eroded from the proximal exposed shelf,
which may have temporarily overprinted the impact of water mass transport changes on the sedimentary Pa/Th ratio.

To summarize, our cross-wavelet transforms and spectrogram analysis results show that changes in water mass transport between ~1300 and 2300 m depth in the western equatorial Atlantic (i.e. within a ~1000 m water layer above MD09-3257 core site) preceded changes in precipitation over the adjacent continent by 110 to 400 y at D-O frequencies, and by 280 to 980 y at Heinrich-like frequencies.
We suggest that the larger lead of ocean circulation changes with respect to tropical South American precipitation changes at Heinrich-like frequencies than at D-O frequencies is likely related to the action of a positive feedback in the case of Heinrich stadials, in agreement with (Burckel et al., 2015). In that case, an AMOC slowdown would lead to sub-surface warming at high northern latitudes, inducing ice-sheet calving and iceberg discharges that would in turn reinforce the initial AMOC slowdown. In contrast, the absence of marked ice rafted detritus layers in North Atlantic sediments during D-O stadials
suggests that in the case of D-O stadials, AMOC slowdowns did not trigger such a positive feedback and hence remained





limited (Burckel et al., 2015). Such a scenario remains to be tested by numerical experiments performed over several thousands of years in glacial conditions with climate models computing water and calcite $\delta^{18}O$, DIC $\delta^{13}C$, and sedimentary Pa/Th.

*Data availability*. Data related to this article are available as a supplement file and on Pangaea.

*Author contribution*. CW and SP designed the research; EB, PB, JL, FT, AD performed the sedimentary Pa/Th measurements; BL performed wavelet analyses; DF and LV contributed expert advices on statistical results and performed spectrogram analyses; LM produced sedimentary Pa/Th values and error bars from MC-ICPMS output. CW, NVR and TD participated in the 2009 RETRO coring cruise. HA contributed expert knowledge on the Brazilian margin. CW and TD
obtained funding. CW and BL wrote the manuscript.

*Competing interests*. The authors declare that they have no conflict of interests.

*Acknowledgements*. This is a contribution to the ACCLIMATE ERC project; the research leading to these results has received funding from the European Research Council under the European Union's Seventh Framework Programme (FP7/2007-2013)/ERC grant agreement n° 339108. Core MD09-3257 was collected on board R/V Marion Dufresne during
the 2009 RETRO coring cruise, supported by IPEV, ANR project ANR-09-BLAN-0347 and the ESF EUROMARC project "RETRO". We thank the IPEV team, crew members of R/V Marion Dufresne and all scientists who participated in the 2009 RETRO cruise. We also thank M. Roy-Barman for advices on Pa/Th measurements on LSCE MC-ICP-MS. We acknowledge C. Moreau, J.-P. Dumoulin, and the UMS ARTEMIS for AMS [14]C dates, as well as G. Isguder, L. Mauclair and F. Dewilde for invaluable technical assistance. This is LSCE contribution 6408.

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

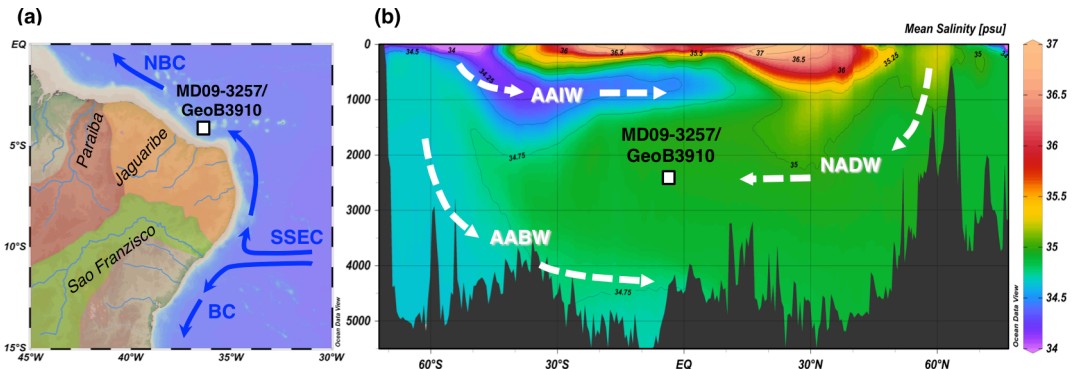

**Figure 1: (a)** Map showing the position of the main Brazilian rivers and surface currents that could influence the terrigenous input at the study site indicated by a white square. The orange area represents the catchment area of the local rivers directly delivering sediments to our study site, and the green area represents the catchment area of the Sao Franzisco River (Milliman et al., 1975). **(b)** Salinity section showing the core site and main water masses in the modern Atlantic Ocean.



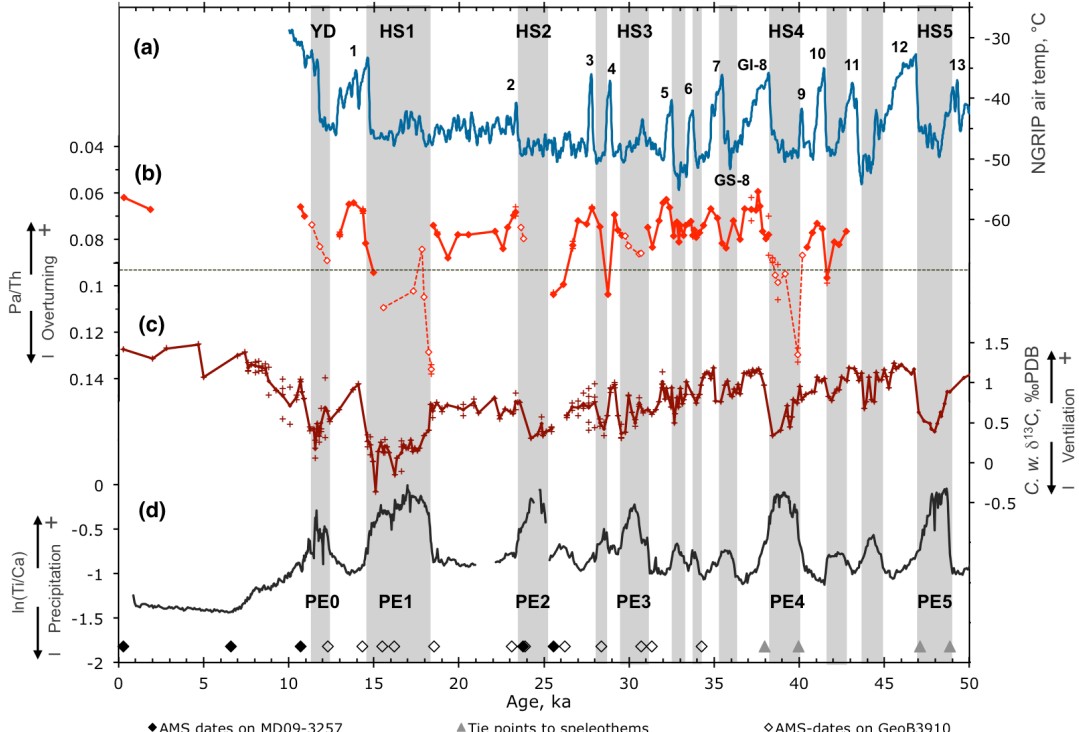

**Figure 2:** MD09-3257 Pa/Th, ln(Ti/Ca) and composite *C. wuellerstorfi* δ¹³C (δ¹³C_Cw) records versus MD09-3257 age scale, independent from NGRIP age scale. **(a)** NGRIP air temperature versus GICC05 age scale (Kindler et al., 2014), transposed from ky b2k to ka. Greenland interstadials are numbered according to (Rasmussen et al., 2014). To avoid over-crowding of the figure, Greenland stadials

5    (GS) and interstadials (GI) are explicitly named only in the case of GS-8 and GI-8. **(b)** Core MD09-3257 Pa/Th record. Empty symbols denote data points that may be affected by terrigenous fluxes and should be interpreted with caution; crosses denote replicate measurements; the red line connects average values (filled symbols). Pa/Th could not be measured over the first half of PE2 because of the occurrence of two small sand layers (Burckel et al., 2015). **(c)** Core MD09-3257 and GeoB3910-2 composite δ¹³C_Cw record (Vazquez Riveiros et al., in prep.); crosses denote replicate measurements. **(d)** MD09-3257 ln(Ti/Ca). Diamonds above the X-axis indicate calibrated

10    radiocarbon dates in MD09-3257 (filled symbols) and GeoB3910-2 (empty symbols). Triangles indicate alignment tie points to South American speleothem δ¹⁸O (Vazquez Riveiros et al., in prep.). Grey bands delineate precipitation events recorded in MD09-3257 ln(Ti/Ca).



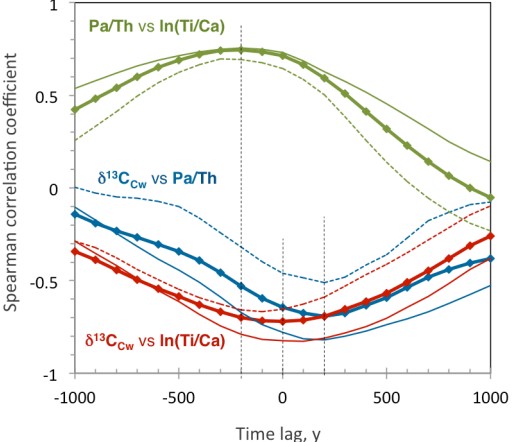

**Figure 3:** Spearman correlation coefficient of $\delta^{13}C_{Cw}$ vs Pa/Th (blue curves), of $\delta^{13}C_{Cw}$ vs ln(Ti/Ca) (red curves), and of Pa/Th vs ln(Ti/Ca) (green curves), as a function of the time lag. A positive time lag means that series 1 lags series 2 (e.g. $\delta^{13}C_{Cw}$ lags Pa/Th); a negative time lag means that series 1 leads series 2 (e.g. Pa/Th leads ln(Ti/Ca)). Bold lines correspond to the calculation over the entire time interval 10.6-42.6 ka, thin lines to the calculation over 10.6-26.6 ka, and thin dashed lines to the calculation over 26.6-42.6 ka. Vertical dashed lines indicate the time lags corresponding to the maximum correlation coefficients for the three pairs of series over the entire time interval.




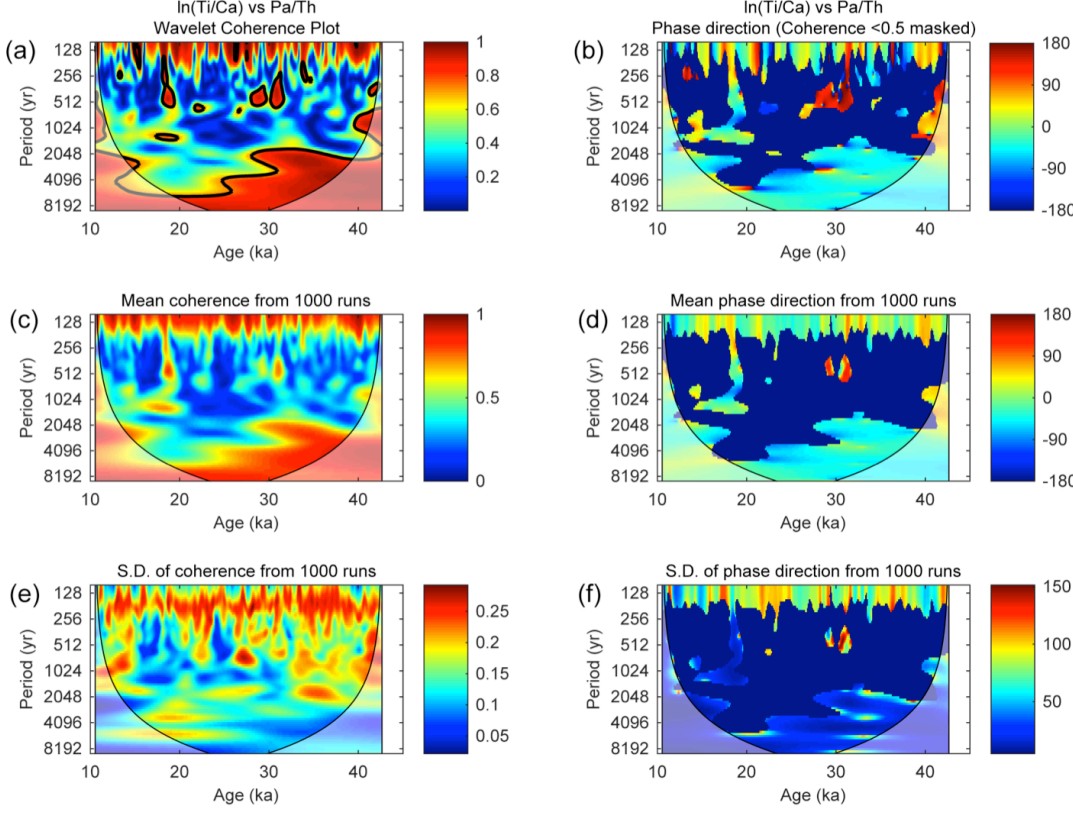

**Figure 4:** Cross-wavelet transform of MD09-3257 ln(Ti/Ca) versus Pa/Th. **(a, b)** Wavelet coherence and phase direction computed using (Grinsted et al., 2004) software. The thick contour line corresponds to the 95% confidence level against red noise. Phase direction is computed only for coherences higher than 0.5. **(c, d)** Mean coherence and phase direction computed out of 1000 Monte Carlo simulations. **(e, f)** Standard deviation around the mean coherence and phase direction computed out of these 1000 Monte Carlo simulations.



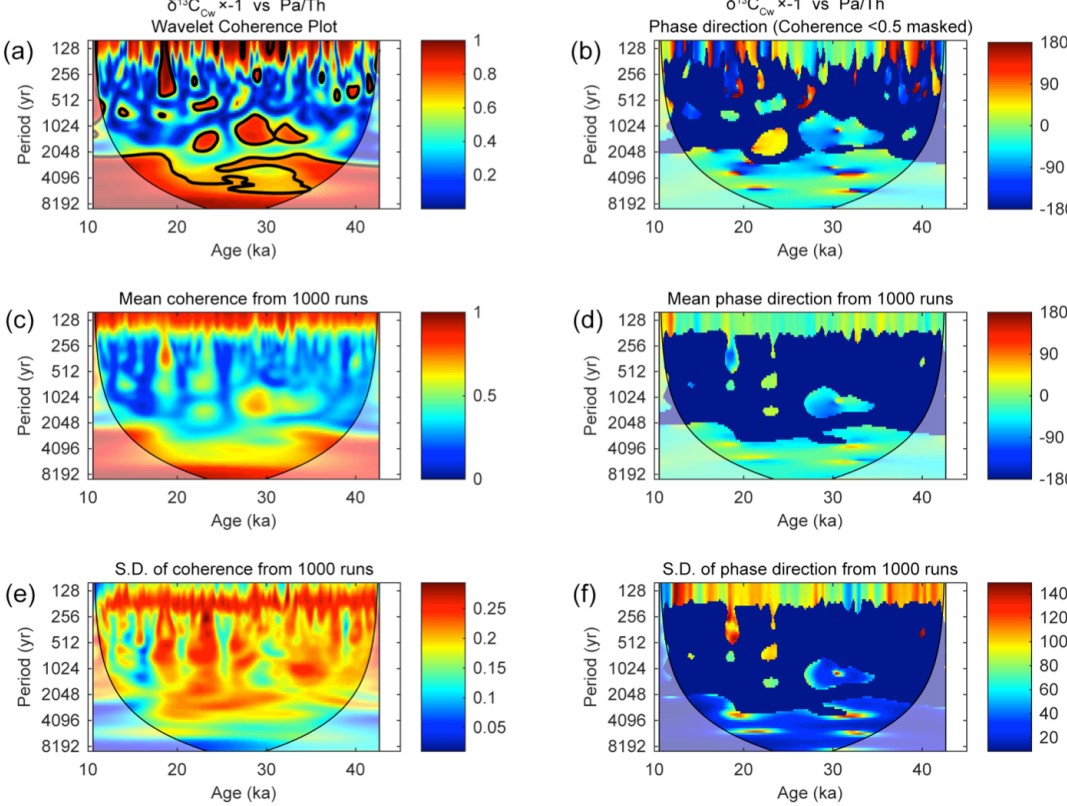

**Figure 5:** Cross-wavelet transform of $\delta^{13}C_{Cw}$ composite record versus MD09-3257 Pa/Th. $\delta^{13}C_{Cw}$ values have been multiplied by (-1) to allow a straightforward reading of the relative phase between a decrease in $\delta^{13}C$ and an increase in Pa/Th. **(a-f)** as in Fig. 4.



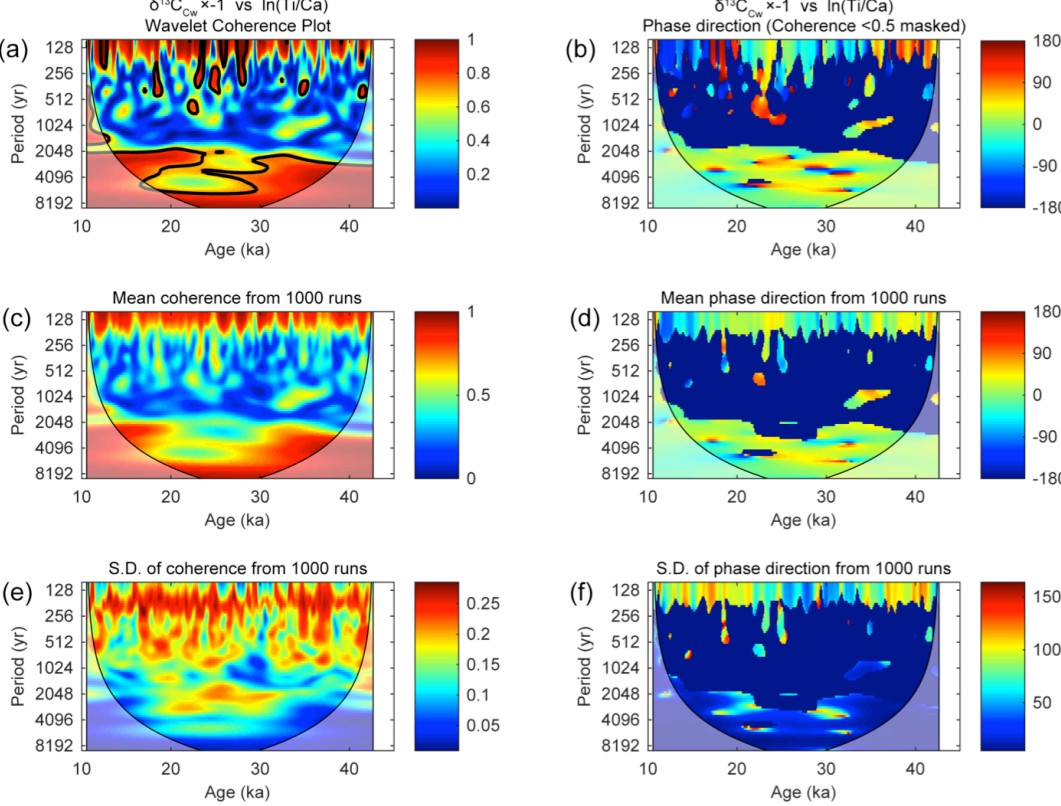

**Figure 6:** Cross-wavelet transform of $\delta^{13}C_{Cw}$ composite record versus MD09-3257 ln(Ti/Ca). $\delta^{13}C_{Cw}$ values have been multiplied by (-1) to allow a straightforward reading of the relative phase between a decrease in $\delta^{13}C$ and an increase in ln(Ti/Ca). **(a-f)** as in Fig. 4 and 5.



**Table 1.** Relative phases over regions of the cross-wavelet graphs corresponding to coherences > 0.5

| | Time interval | Period range | Perio- dicity | Phase, deg | 1σ, deg | Phase, y | 1σ, y | Comment |
|---|---|---|---|---|---|---|---|---|
| **ln(Ti/Ca) vs Pa/Th (Fig. 4)** | 28-40 ka* | 1000-3000 | **2000** | **-46.7** | **25.2** | **-259** | **140** | **Pa/Th leads ln(Ti/Ca)** |
| | 15-40 ka* | 4000-6000 | **5000** | **-45.4** | **24.8** | **-631** | **345** | **Pa/Th leads ln(Ti/Ca)** |
| **d$^{13}$C vs Pa/Th (Fig. 5)** | 28-40 ka | 1000-3000 | **2000** | **-50.2** | **39.7** | **-279** | **244** | **Pa/Th leads d13C** |
| | 15-40 ka | 4000-6000 | 5000 | -14.1 | 37.1 | -196 | 525 | not significant |
| **d13C vs ln(Ti/Ca) (Fig. 6)** | 28-40 ka | 1000-3000 | 2000 | 17 | 24.3 | 94 | 171 | not significant |
| | 15-40 ka | 4000-6000 | 5000 | 10.8 | 42.9 | 150 | 606 | in phase |

*within these time intervals, only results from the unshaded region of the wavelet graphs are taken into account.