# Peer review of "Relative timing of precipitation and ocean circulation changes in the western equatorial Atlantic over the last 45 ky"

_Climate of the Past, 2018_

## Referee Comment (RC1) · Anonymous Referee #1 · 20 Apr 2018

General Comments: Waelbroeck et al. present results for 2 cores from the North Brazilian margin, using proxies for AMOC-related ocean circulation changes (Pa/Th, C. wuellerstorfi d13C) and South American precipitation events (Ti/Ca). As the proxy records were generated from the same location/core (more or less) the authors argue that there are no lead/lags related to age model uncertainty and hence this allows to properly assess the phase relationship between AMOC and South American rainfall during the last 45 kyr. Their new data allows to not only focus on the last 4 Heinrich stadials (already presented in Burckel et al., 2015), but also D-O events with shorter frequencies. Based on the careful analysis of their data (using mainly cross-wavelet analyses), they infer that changes in water mass transport in the mid-depth range of the

western equatorial Atlantic precede precipitation changes in Brazil. This is especially the case at Heinrich-like frequencies, and less so at D-O frequencies, which they relate to a positive feedback mechanism in the ocean/atmosphere system during Heinrich stadials.

The manuscript is well written, well structured, and concerns an important topic that is certainly relevant for Climate of the Past. In essence, this paper is an evolution of the Burckel et al. (2015) paper, but with some extra Pa/Th and d13C data, which makes it possible to better study changes in water mass transport over Dansgaard-Oeschger frequencies. In general, the authors carefully address the possible biases on Pa/Th and other proxy records (influences by marine productivity, differential bioturbation, currents etc.) and deliver quite a good case for the ocean circulation changes and leads/lags to South American precipitation during D-O/Heinrich stadials. I do have some reservations about the age model, as I think there are some details missing in text to properly evaluate the chronology (and uncertainty). Moreover, more details on some of the geochemical analyses are required (citing an "in prep." paper is in my opinion not enough). If these two main issues are properly addressed, I certainly recommend publication in Climate of the Past.

Specific Comments: p.2 line 18: XRF, do you mean XRF core scanning (as in Jaeschke et al., 2007, done with the CORTEX scanner) or with more conventional XRF done on glass beads/pressed tablets? If it is the former, please change the abbreviation throughout the text, e.g., XRF-core-scanning (XCS).

p. 3, line 9: I miss a paragraph on the geochemical measurements performed to derive the Ti/Ca ratio. The Ti/Ca values were already published in Burckel et al. (2015), but I cannot find the XRF methods in there (I could have overlooked it). The best would be to give details on the used methods here, at least briefly. Note also that if you used the same method as Jaeschke et al. (2007), you probably used a different core scanner (Avaatech? Itrax?).

p.3, line 22: Log-ratios of Ti/Ca are indeed the way to go, also, because they allow a better statistical modelling of compositional data (see Weltje and Tjallingii, 2008; normal ratios are asymmetric). It would be good to shortly address this too in this sentence.

p.3, lines 9-27 (Chronology): I find the chronological section not yet satisfying. For instance, I miss what software was used to calculate the age model (OxCal?), and more technical details (reservoir age? uncertainties?). I see that in Burckel et al. (2015) the age model is addressed in one of the 16 Supplements of that paper, but I think it is important to at least briefly address the most important parts again. As written now, you might as well have used a simple linear model between the age points, but I cannot find that in the text. As the age model is clearly crucial for the results of this paper, the details should be better outlined (and not simply covered by a reference to an "in prep." paper). For instance, did the authors use the state-of-the-art OxCal Bayesian modeling, and if not, why not? The authors should read the Sections 3 and 4 in the Supplement of Grant et al. (2012), who do a good job of obtaining the chronological uncertainties with a Bayesian deposition model in OxCal (also to calculate lags/leads between proxy records, albeit with a different scope).

p.4, lines 1-16: The details on the d13C methods should be given here, and not in the Vazquez Riveiros (in prep.) paper.

p.4-5 (New sedimentary Pa/Th data): How was the discrete sampling performed for Pa/Th? This might be important for direct comparison of Pa/Th to Ti/Ca (from XRF-scanning?) during the D-O variability. Are the analyses by both methods exactly performed on the same sediment intervals? Core scanner intervals are often deviating from those that are discretely sampled (e.g., at 1-cm resolution, a measurement at 5 cm is covering the interval from 4.5 to 5.5cm, with an Avaatech core scanner). This might somewhat influence the lead/lag calculations, and may require resampling of the Ti/Ca data (although the impact is probably small).

[Figure]

p. 5, lines 19-20: Add shortly why the 232Th is indicative of the vertical terrigenous flux (detrital origin?).

p.10, lines 8-12: The uncertainty of the leads and lags in the cross-wavelets should already be given in the Methods (section "cross-correlation and wavelet analysis"). The error propagation is not entirely clear to me, did the authors use a mean squared error (MSE)?

p. 11, line 23: Is the cross-correlation really imprecise and unreliable, or does it just lump all frequency signals into one and give you an average output, which is basically correct for the time window that was analyzed? The authors could have used different time windows (e.g., 3000 years and 6000 years) and calculate a running correlation across the whole interval (with one of the records shifted towards the other in different time steps). The result from such a running correlation test will/would be probably very similar to the cross-wavelet analyses. Cross-correlation is not imprecise or unreliable, just not the most suitable method to study non-stationary climate signals. I suggest to change this sentence, and also that at p. 11 line 29, more focusing on the fact that these cross-correlations cannot be used to disentangle the leads/lags of variable frequencies in the proxy records.

p. 12, line 26: What is the reason to not just use a ln(K/Ca) ratio, instead of ln(Ti/Ca), to circumvent these problems? (Other than the reason that previous studies used Ti/Ca, but probably did not consider these bioturbation effects).

p.13, line 10: To me it seems that for HS3 there is also not a clear visible lead of Pa/Th relative to ln(Ti/Ca). Is this not what you expect considering that the origin of icebergs/IRD seems to be more European orientated for HS3 (Gwiazda et al., 1996; Henry et al., 2016), while the others find their origin mainly from the Laurentide ice sheet? The reduction in overturning seems to be also much less during HS3 compared to the others.

p.13, line 20: Doesn't the 232Th flux show that the vertical terrigenous flux was largest

during HS4?

p.14, line 27: Is a 2-4cm downward shift also plausible for differential bioturbation? I suppose there is always bioturbation of both fine and coarse particles.

Technical Comments: p.2 line 8-10 ("In the best. . .into calendar ages"): Sentence does not read well. Rephrase/break up sentence.

p.2 line 26: Add when the core was recovered.

p. 3, line 27: Please write "as defined by Rasmussen et al. (2014)". This should also be done for the other parts of the text where citations are part of the sentence.

p. 8, line 21: Table S2 considers the opal measurements, which Table needs to be referred to?

Figure 1: I think a larger overview map of South America would have been nice here (e.g., Burckel et al., 2015)

Figure S1: Multiplier for ln(Ti/Ca), is this really necessary? Is it not sufficient to change the range on the y-axis?

Figure S1: The unit for the sedimentation rate is missing partially on the y-axis.

References: Burckel, P., Waelbroeck, C., Gherardi, J.-M., Pichat, S., Arz, H., Lippold, J., Dokken, T., and Thil, F.: Atlantic Ocean circulation changes preceded millennial tropical South America rainfall events during the last glacial, Geophys. Res. Lett., 42, 411-418, 2015.

Grant, K.M., Rohling, E.J., Bar-Matthews, M., Ayalon, A., Medina-Elizalde, M., Ramsey, C.B., Satow, C., Roberts, A.P.: Rapid coupling between ice volume and polar temperature over the past 150,000 years. Nature 491, 744-747, 2012.

Gwiazda, R.H., Hemming, S.R., Broecker, W.S.: Provenance of icebergs during Heinrich event 3 and the contrast to their sources during other Heinrich episodes, Paleoceanography, 11(4), 371-378, 1996.

Henry, L., McManus, J. F., Curry, W. B., Roberts, N. L., Piotrowski, A. M., and Keigwin, L. D.: North Atlantic ocean circulation and abrupt climate change during the last glaciation, Science, 353, 470-474, 2016.

Jaeschke, A., Rühlemann, C., Arz, H., Heil, G., and Lohmann, G.: Coupling of millennial-scale changes in sea surface temperature and precipitation off northeastern Brazil with high-latitude climate shifts during the last glacial period, 20 Paleoceanography, 22, PA4206, 2007.

Weltje, G.J., Tjallingii, R.: Calibration of XRF core scanners for quantitative geochemical logging of sediment cores: theory and application. Earth and Planetary Science Letters 274, 423-438, 2008.

---

## Referee Comment (RC2) · R. Francois (Referee) · 15 May 2018

Waelbroeck et al. have measured Ti/Ca and Pa/Th in a core taken on the margin of northern Brazil, which records rainfall on the nearby continental area and the strength of the Atlantic meridional overturning circulation at intermediate depths. Since the two measurements are done on the same core, they are able to determine the phase relationship between the two variables with minimum uncertainty, which gives them new insights into the response of the ITCZ to changes in the AMOC. They use wavelet analysis to determine phase lags at different frequencies and show that ITCZ movement lags changes in AMOC both at the D/O and HE frequency, but more so at the HE frequency. They attribute this difference to a positive feedback between the strength of the AMOC, seawater temperature and iceberg discharge, which they had first proposed in an earlier publication. The authors pay due attention to the possible impact of bioturbation on the phase relationship between Ti/Ca and Pa/Th, which they convincingly rule out, and to multiple caveats in the interpretation of Pa/Th in term of circulation changes. The paper is clearly written and provides important new findings. I recommend publication after considering the relatively minor comments below (note, however, that I am unable to provide knowledgeable comments on the technicalities of wavelet analysis).

While the authors have clearly established the lag between Ti/Ca and Pa/Th, their ultimate goal is to establish the lag in the response of the ITCZ to changes in AMOC. I think the authors should also discuss the extent to which there might be lags between processes and proxies. For instance, would a change in AMOC translate instantaneously into a change in Pa/Th in their core? I think this is unlikely. Pa/Th recorded in sediments is controlled by the ratio between lateral transport by circulation and vertical transport by scavenging of the Th and Pa produced in the water column.  Even if seawater were flowing from the north Atlantic to the Brazilian margin through a pipe (i.e. changes in deep water formation would translate into an instantaneous change in lateral velocity in the pipe), there should still be a lag between sediment Pa/Th and changes in AMOC, depending on the response time of dissolved Th and Pa in the water column overlying the coring site. While the response time of Th is decadal, the full expression on circulation changes on Pa may take several centuries. In addition, the "pipe" is of course an unrealistic cartoon of the AMOC. In reality, I would expect an additional lag between lateral velocity at the coring site and changes in deep water formation, but at this point this is just intuitive and it is well beyond me to guess how long or how short this lag would be. Nonetheless, I think the authors could bring this up and indicate that the lag between Ti/Ca and Pa/Th should be taken as a minimum of the lag of the response of the ITCZ to changes in AMOC. There might also be a lag between Ti/Ca and the change in the seasonal latitudinal range of position of the ITCZ depending on the location of the region supplying lithogenics to the coring site. For instance, if the region is farther south from the southernmost zone of precipitation before the change in AMOC, it may take more time for the ITCZ to reach this region.

While the authors have taken into account how changes in scavenging could obscure the interpretation of Pa/Th in terms of circulation, I think it would also be worth mentioning that interpreting changes in circulation from a single core can also be problematic. While it is correct that higher rate of AMOC should result in a lower sediment Pa/Th when averaged over an entire ocean basin, that may not be correct for any core. Depending on the proximity of the coring location to the site of deep water formation, decreasing the AMOC may actually decrease sediment Pa/Th (e.g. Luo et al., 2010; Fig. 14). I

would suggest specifying that we would expect to see an increase in Pa/Th with decreasing rate of AMOC at the coring site of this study because it is sufficiently removed from the site of deep water formation.

Accordingly, I would change the wording on line 4-5 p5: "[*when average over an entire ocean basin*], high (low) flow rates therefore result in high (low) Pa export…"

Line 26, p7: shorter stadial may have lower increase in Pa/Th because they were too short to allow the full expression of the increase in Pa/Th (limited by the response time of Pa in the water column).

Line 10, p8: (including Pa/Th values susceptible to be partially impacted by large particles flux [*or boundary scavenging resulting from slower AMOC*])

Line 24, p12: Why not use Th-normalized Ti instead of Ti/Ca to totally eliminate the effect of changes in carbonate dissolution/production?

Line 8; p13: Briefly describe what the "independent approach" is.

---

## Author Comment (AC1) · 28 Jun 2018

**Detailed response to Referee 1's comments**

We are very grateful for the constructive and helpful comments we received from both reviewers. Accounting for them has been of great help to improve the manuscript.

General Comments: Waelbroeck et al. present results for 2 cores from the North Brazilian margin, using proxies for AMOC-related ocean circulation changes (Pa/Th, C. wuellerstorfi d13C) and South American precipitation events (Ti/Ca). As the proxy records were generated from the same location/core (more or less) the authors argue that there are no lead/lags related to age model uncertainty and hence this allows to properly assess the phase relationship between AMOC and South American rainfall during the last 45 kyr. Their new data allows to not only focus on the last 4 Heinrich stadials (already presented in Burckel et al., 2015), but also D-O events with shorter frequencies. Based on the careful analysis of their data (using mainly cross-wavelet analyses), they infer that changes in water mass transport in the mid-depth range of the western equatorial Atlantic precede precipitation changes in Brazil. This is especially the case at Heinrich-like frequencies, and less so at D-O frequencies, which they relate to a positive feedback mechanism in the ocean/atmosphere system during Heinrich stadials.

The manuscript is well written, well structured, and concerns an important topic that is certainly relevant for Climate of the Past. In essence, this paper is an evolution of the Burckel et al. (2015) paper, but with some extra Pa/Th and d13C data, which makes it possible to better study changes in water mass transport over Dansgaard-Oeschger frequencies. In general, the authors carefully address the possible biases on Pa/Th and other proxy records (influences by marine productivity, differential bioturbation, currents etc.) and deliver quite a good case for the ocean circulation changes and leads/lags to South American precipitation during D-O/Heinrich stadials. I do have some reservations about the age model, as I think there are some details missing in text to properly evaluate the chronology (and uncertainty). Moreover, more details on some of the geochemical analyses are required (citing an "in prep." paper is in my opinion not enough). If these two main issues are properly addressed, I certainly recommend publication in Climate of the Past.

We have added all the requested information concerning the age model and isotopic analyses in the material and methods section, as described in details below.

Specific Comments: p.2 line 18: XRF, do you mean XRF core scanning (as in Jaeschke et al., 2007, done with the CORTEX scanner) or with more conventional XRF done on glass beads/pressed tablets? If it is the former, please change the abbreviation throughout the text, e.g., XRF-core-scanning (XCS).

We mean XRF core scanning, as in Jaeschke et al. (2007). The XRF data of core MD09-3257 were produced with an AVAATECH XRF core scanner, as now described in the material and methods section of the article. We prefer to keep the abbreviation XRF throughout the text though because this is the abbreviation commonly used and the abbreviation most easily understandable by the reader since it directly refers to the physical principal behind the measurement technique. Moreover, we use the GeoB3910 XRF data from Jaeschke et al. (2007), who used the denomination XRF throughout their paper. Also, thanks to the new paragraph describing the measurement method, there can be no confusion any longer.

p. 3, line 9: I miss a paragraph on the geochemical measurements performed to derive the Ti/Ca ratio. The Ti/Ca values were already published in Burckel et al. (2015), but I cannot find the XRF methods in there (I could have overlooked it). The best would be to give details on the used methods here, at least briefly. Note also that if you used the same method as Jaeschke et al. (2007), you probably used a different core scanner (Avaatech? Itrax?).

We thank Referee 1 for having identified this omission in the submitted version of our article. We have added the following paragraph to the material and methods section:

"*X-Ray Fluorescence Spectrometry*

Elemental composition was measured employing nondestructive, profiling X-ray fluorescence (XRF) spectrometry. The measurements were made using an AVAATECH XRF Core Scanner at the Bjerkness Centre for Climate Research, Bergen (Norway) at intervals of 0.5 mm on core MD09-3257, and using a CORTEX XRF Scanner at the Bremen Integrated Ocean Drilling Program core repository at intervals of 0.4 cm on core GeoB3910-2 (Jaeschke et al., 2007). This automated scanning method allows for a rapid qualitative determination of the geochemical composition of the sediment at very high resolution (Croudace and Rothwell, 2015)."

p.3, line 22: Log-ratios of Ti/Ca are indeed the way to go, also, because they allow a better statistical modelling of compositional data (see Weltje and Tjallingii, 2008; normal ratios are asymmetric). It would be good to shortly address this too in this sentence.

We thank Referee 1 for his/her remark and for his/her recommendation to read the article Weltje and Tjallingii (2008), which we found very informative. We have changed the sentence

"Here, we use XRF ln(Ti/Ca) rather than Ti/Ca because small precipitation events are more clearly marked in ln(Ti/Ca) than in Ti/Ca."

into

"Here, we use XRF ln(Ti/Ca) rather than Ti/Ca because log-ratios provide a unique measure of sediment composition, in contrast to simple ratios which are asymmetric (i.e. conclusions based on evaluation of A/B cannot be directly translated into equivalent statements about B/A) and hence suffer from statistical intractability (Weltje and Tjallingii, 2008)."

p.3, lines 9-27 (Chronology): I find the chronological section not yet satisfying. For instance, I miss what software was used to calculate the age model (OxCal?), and more technical details (reservoir age? uncertainties?). I see that in Burckel et al. (2015) the age model is addressed in one of the 16 Supplements of that paper, but I think it is important to at least briefly address the most important parts again. As written now, you might as well have used a simple linear model between the age points, but I cannot find that in the text. As the age model is clearly crucial for the results of this paper, the details should be better outlined (and not simply covered by a reference to an "in prep." paper). For instance, did the authors use the state-of-the-art OxCal Bayesian modeling, and if not, why not? The authors should read the Sections 3 and 4 in the Supplement of Grant et al. (2012), who do a good job of obtaining the chronological uncertainties with a Bayesian deposition model in OxCal (also to calculate lags/leads between proxy records, albeit with a different scope).

All radiocarbon dates were converted to calendar dates using the OxCal 4.2 software, the IntCal13 calibration curve, and a surface water reservoir age of $550 \pm 50$ y between $0 - 18$ ka

(Key et al., 2004), and of 750 ± 250 y between 18 – 31 ka (Freeman et al., 2016). The final age models were obtained using the state-of-the-art OxCal Bayesian modeling. We have added the age uncertainty for each core depth in Table S1 and S2.

The article (Vazquez Riveiros et al., in prep.) is unfortunately not accepted yet. We have thus added all the information requested by Referee 1 to the section describing the chronology of our cores. The following sentences have been added:

"The chronology of core GeoB3910-2 is based on 17 monospecific radiocarbon dates between 0 – 31 ky (Burckel et al., 2015; Jaeschke et al., 2007). The Ti/Ca record of core GeoB3910-2 was aligned to that of core MD09-3257 in order to transfer the radiocarbon dates of GeoB3910-2 between 12–36 ka to this nearby core. In addition, five monospecific radiocarbon dates between 1–21 ka were obtained directly on core MD09-3257. Speleothem tie points were used to derive the chronology of this core between 38–48 ka (Table S1 and S2) (Vazquez Riveiros et al., submitted). All radiocarbon dates were converted to calendar dates using the OxCal 4.2 software, the IntCal13 calibration curve (Reimer et al., 2013), a surface water reservoir age of 550 ± 50 y between 0–18 ka (Key et al., 2004), and of 750 ± 250 y between 18–31 ka (Freeman et al., 2016). The final age models of cores GeoB3910-2 and MD09-3257 were obtained using a *P_Sequence* depositional model (Bronk Ramsey, 2008), i.e. a Bayesian algorithm producing posterior probability distributions for each core depth (Table S1 and S2) (Vazquez Riveiros et al., submitted)."

p.4, lines 1-16: The details on the d13C methods should be given here, and not in the Vazquez Riveiros (in prep.) paper.

We have added the requested information to the material and methods section:

"Epifaunal benthic foraminifers of the *Cibicides wuellerstorfi* species were handpicked in the >150 mm size fraction (Vazquez Riveiros et al., submitted). Core MD09-3257 *C. wuellerstorfi* $^{13}C/^{12}C$ ($\delta^{13}C$, expressed in ‰ versus Vienna Pee-Dee Belemnite, VPDB) was measured at the LSCE on Finnigan $\Delta$+ and Elementar Isoprime mass spectrometers on samples of 1 to 3 specimens. VPDB is defined with respect to NBS-19 calcite standard ($\delta^{18}O$ = -2.20 ‰ and $\delta^{13}C$ = +1.95 ‰). The mean external reproducibility (1$\sigma$) of carbonate standards is ± 0.03 ‰ for $\delta^{13}C$; measured NBS-18 $\delta^{18}O$ is -23.27 ± 0.10 and $\delta^{13}C$ is -5.01 ± 0.03 ‰ VPDB. Core GeoB3910-2 *C. wuellerstorfi* $\delta^{13}C$ was measured at the University of Bremen, Germany, on a Finnigan MAT 252 mass spectrometer equipped with an automatic carbonate preparation device on samples of 1 to 5 specimens, with a mean external reproducibility (1$\sigma$) for carbonate standards of ± 0.05 ‰ for $\delta^{13}C$."

p.4-5 (New sedimentary Pa/Th data): How was the discrete sampling performed for Pa/Th? This might be important for direct comparison of Pa/Th to Ti/Ca (from XRFscanning?) during the D-O variability. Are the analyses by both methods exactly performed on the same sediment intervals? Core scanner intervals are often deviating from those that are discretely sampled (e.g., at 1-cm resolution, a measurement at 5 cm is covering the interval from 4.5 to 5.5cm, with an Avaatech core scanner). This might somewhat influence the lead/lag calculations, and may require resampling of the Ti/Ca data (although the impact is probably small).

Pa/Th was measured on discrete 1cm-thick samples, as well as *C. wuellerstorfi* $\delta^{13}C$. In contrast, the use of an Avaatech XRF Core Scanner to measure MD09-3257 elemental ratios allowed us to produce a quasi-continuous MD09-3257 Ti/Ca signal with 1 measurement every

0.5 mm. For the purposes of the present study, we first resampled and dated core MD09-3257 Ti/Ca signal at depth intervals of 0.5 cm. Then, prior to time series analyses, as explained p. 6, lines 7-8, we resampled all three studied time series with constant time steps varying between 50 and 500 y (corresponding to ~0.5 to 5 cm spacing, knowing that the mean sedimentation rate is 10 cm/ky). Therefore, the fact that the initial sample thickness of Pa/Th or *C. wuellerstorfi* $\delta^{13}$C measurements is different from that of Ti/Ca measurements, has no impact on the lead/lag calculations.

p. 5, lines 19-20: Add shortly why the 232Th is indicative of the vertical terrigenous flux (detrital origin?).

We have modified the sentence

"The $^{230}$Th-normalized $^{232}$Th flux, hereafter simply referred to as the $^{232}$Th flux, is indicative of the vertical terrigenous flux to the core site."

into

"The $^{230}$Th-normalized $^{232}$Th flux, hereafter simply referred to as the $^{232}$Th flux, is indicative of the vertical flux of terrigenous material at the core site, since $^{232}$Th is a trace element that is mostly contained in the continental crust (Taylor and McLennan, 1985) and is thus commonly used as a geochemical tracer for material of detrital origin (*e.g.* Anderson et al. (2006))."

p.10, lines 8-12: The uncertainty of the leads and lags in the cross-wavelets should already be given in the Methods (section "cross-correlation and wavelet analysis"). The error propagation is not entirely clear to me, did the authors use a mean squared error (MSE)?

We computed the uncertainty of the leads and lags produced by the wavelet analysis assuming Gaussian error propagation of the two independent uncertainties described p.10, lines 8-12. We computed the total 1σ uncertainty as the square root of the sum of the different variances representing the different sources of uncertainty taken into account. We have clarified this and changed the sentence

"Note that uncertainties for leads and lags given in Table 1 are computed as the propagation of two uncertainties: (i) […] (Fig. 4-6d), and (ii) […] (Fig. 4-6f)."

into

"The uncertainties of the leads and lags (Table 1) are computed assuming Gaussian error propagation of the two following independent uncertainties: (i) […] (Fig. 4-6d), and (ii) […] (Fig. 4-6f)."

However, we cannot move this paragraph from the results section to the methods section because the description of the two independent sources of uncertainty involves the description of the wavelet results given in Figure 4.

p. 11, line 23: Is the cross-correlation really imprecise and unreliable, or does it just lump all frequency signals into one and give you an average output, which is basically correct for the time window that was analyzed? The authors could have used different time windows (e.g., 3000 years and 6000 years) and calculate a running correlation across the whole interval (with one of the records shifted towards the other in different time steps). The result from such a running correlation test will/would be probably very similar to the cross-wavelet analyses. Cross-correlation is not imprecise or unreliable, just not the most suitable method to study

non-stationary climate signals. I suggest to change this sentence, and also that at p. 11 line 29, more focusing on the fact that these cross-correlations cannot be used to disentangle the leads/lags of variable frequencies in the proxy records.

Cross-correlation does indeed lump all frequencies. This method thus yields one unique relative phase for the entire studies record, which is meaningless because different portions of the studied records are characterized by different frequencies. Also, as described p. 6 (lines 4-11), cross correlation consists in computing the correlation coefficient between two time series, after having shifted one with respect to the other by increments of the time step (R script given in the supplementary material). The results of this operation are given in Figure 2 and are not similar to the cross-wavelet analyses for the reason given above.

However, we agree that the sentences p. 11, line 23 and line 29 could be improved and that we should insist on the fact that cross-correlation is not a suitable method to study non-stationary climate signals. We have thus modified

"[…] confirms that the latter method yields imprecise and unreliable results when applied to climatic signals"

into

"[…] confirms that the latter method yields imprecise and unreliable results when applied to non-stationary climatic signals",

and

"However, as shown here, cross-correlation does not yield reliable results when applied to climatic signals of the last glacial."

into

"However, as shown here, cross-correlation is not a suitable method to analyze non-stationary climatic signals such as those of the last glacial."

p. 12, line 26: What is the reason to not just use a ln(K/Ca) ratio, instead of ln(Ti/Ca), to circumvent these problems? (Other than the reason that previous studies used Ti/Ca, but probably did not consider these bioturbation effects).

We agree with Referee 1 that it seems judicious to use ln(K/Ca) or ln(Rb/Ca) instead of ln(Ti/Ca) as a proxy of runoff from the adjacent continent. However, the present study builds on previous studies from the same region and cores using ln(Ti/Ca) or Ti/Ca, so we chose to use ln(Ti/Ca) and simply verified if an offset between ln(K/Ca) or ln(Rb/Ca) and ln(Ti/Ca) was detectable.

p.13, line 10: To me it seems that for HS3 there is also not a clear visible lead of Pa/Th relative to ln(Ti/Ca). Is this not what you expect considering that the origin of icebergs/IRD seems to be more European orientated for HS3 (Gwiazda et al., 1996; Henry et al., 2016), while the others find their origin mainly from the Laurentide ice sheet? The reduction in overturning seems to be also much less during HS3 compared to the others.

We agree that the reduction in overturning as recorded by Pa/Th is much smaller during HS3 compared to the other Heinrich stadials. Concerning the lead of Pa/Th relative to ln(Ti/Ca) marking the beginning of HS3, the only notable difference between that transition towards higher Pa/Th values and the transitions corresponding to the other Heinrich stadials, is one single Pa/Th data point (dated at ~31 ka) which was not duplicated and makes the transition a

little noisy. We thus prefer not to draw firm conclusions from the presence of this single data point.

p.13, line 20: Doesn't the 232Th flux show that the vertical terrigenous flux was largest during HS4?

We thank Referee 1 for this question that led us to realize that the discussion concerning the different phasing observed for Heinrich Stadial 1 than for the Younger Dryas and the other Heinrich stadials had to be modified.

The large $^{232}$Th flux recorded during both HS4 and HS1, together with the similarity in Pa/Th and Ti/Ca amplitudes during these two Heinrich stadials indicate that the different phasing observed for Heinrich Stadial 1 is most likely not due to a difference in terrestrial input.

We have thus replaced this portion of the discussion by the following few sentences:

"Such a different sequence of events seems to indicate that in the case of HS1, the increase in rainfall over tropical South America during HS1 was not a response to a decrease in Atlantic overturning circulation. Instead, a southward shift of the low-latitude atmospheric convection zone (Intertropical Convergence Zone, ITCZ), along with its associated maximum in precipitation, could have occured in response to extended northern high-latitude ice sheets and sea ice cover without any change in ocean circulation (Chiang et al., 2003). This atmospheric mechanism would have prevailed at the beginning of HS1 because ice sheets reached their maximum extent around that time."

Similarly, we removed the two sentences on this topic from our conclusions and modified the conclusions last sentence into:

"Finally, the relative lead of Pa/Th over ln(Ti/Ca) is visible for the YD and for all Heinrich stadials, except HS1. In the case of HS1, the southward shift of the ITCZ could have been an atmospheric response to the maximum extent in northern high-latitude ice sheets and sea ice cover (Chiang et al., 2003) around that time, rather than a progressive response to a slowdown of the AMOC, as is the case of the other stadials. These different atmospheric and oceanic scenarios remain to be tested by numerical experiments performed over several thousands of years in glacial conditions, whereby climate models compute water and calcite $\delta^{18}$O, DIC $\delta^{13}$C, and sedimentary Pa/Th."

We are grateful to Referee 1 for noticing that the discussion of this aspect of our data in the submitted version of our manuscript was not convincing. We are glad that the revised version is improved in this regard.

p.14, line 27: Is a 2-4cm downward shift also plausible for differential bioturbation? I suppose there is always bioturbation of both fine and coarse particles.

The sentence p.14, line 25 to 27 does indeed concern differential bioturbation. We have clarified this by replacing "bioturbation" by "differential bioturbation".

Technical Comments:

We thank Referee 1 for all his/her comments and advices, not only on the article content but also on its form.

p.2 line 8-10 ("In the best. . .into calendar ages"): Sentence does not read well. Rephrase/break up sentence.

We have replaced the very long sentence

"In the best cases, when marine cores are radiocarbon dated, past surface reservoir ages do not vary too much through time, and bioturbation biases remain limited (e.g. for high sedimentation rates), dating uncertainties mainly derive from the calibration of radiocarbon ages into calendar ages."

by 2 sentences:

"When marine cores are radiocarbon dated, uncertainties can arise from bioturbation biases (e.g. Lougheed et al. (2018)) and changes in past surface reservoir ages (Waelbroeck et al., 2001; Thornalley et al., 2011). In the best cases, when changes in past surface reservoir ages and bioturbation biases remain limited, dating uncertainties mainly derive from the calibration of radiocarbon ages into calendar ages."

p.2 line 26: Add when the core was recovered.

Done

p. 3, line 27: Please write "as defined by Rasmussen et al. (2014)". This should also be done for the other parts of the text where citations are part of the sentence.

Done

p. 8, line 21: Table S2 considers the opal measurements, which Table needs to be referred to?

We thank Referee 1 for having noted this omission. We have added a table (Table S4 in the new numbering) containing the cross-correlation results to the supplementary material.

Figure 1: I think a larger overview map of South America would have been nice here (e.g., Burckel et al., 2015)

We chose this degree of zoom in order to be able to clearly represent the different catchment areas. The rationale behind this choice is to provide the reader with the information on the surface currents and Brazilian rivers that may impact on the terrigenous input at the core site.

Figure S1: Multiplier for ln(Ti/Ca), is this really necessary? Is it not sufficient to change the range on the y-axis?

We opted for that solution for simplicity. Importantly, the scaling by 0.3 of the ln(Ti/Ca) of both cores has no incidence on this supplementary figure showing the alignment of GeoB3910-2 ln(Ti/Ca) to MD09-3257 ln(Ti/Ca).

Figure S1: The unit for the sedimentation rate is missing partially on the y-axis.

Fixed!

References: Burckel, P., Waelbroeck, C., Gherardi, J.-M., Pichat, S., Arz, H., Lippold, J., Dokken, T., and Thil, F.: Atlantic Ocean circulation changes preceded millennial tropical South America rainfall events during the last glacial, Geophys. Res. Lett., 42, 411-418, 2015.

Grant, K.M., Rohling, E.J., Bar-Matthews, M., Ayalon, A., Medina-Elizalde, M., Ramsey, C.B., Satow, C., Roberts, A.P.: Rapid coupling between ice volume and polar temperature over the past 150,000 years. Nature 491, 744-747, 2012.

Gwiazda, R.H., Hemming, S.R., Broecker, W.S.: Provenance of icebergs during Heinrich event 3 and the contrast to their sources during other Heinrich episodes, Paleo- ceanography, 11(4), 371-378, 1996.

Henry, L., McManus, J. F., Curry, W. B., Roberts, N. L., Piotrowski, A. M., and Keigwin, L. D.: North Atlantic ocean circulation and abrupt climate change during the last glaciation, Science, 353, 470-474, 2016.

Jaeschke, A., Rühlemann, C., Arz, H., Heil, G., and Lohmann, G.: Coupling of millennial-scale changes in sea surface temperature and precipitation off northeastern Brazil with high-latitude climate shifts during the last glacial period, 20 Paleoceanography, 22, PA4206, 2007.

Weltje, G.J., Tjallingii, R.: Calibration of XRF core scanners for quantitative geochemical logging of sediment cores: theory and application. Earth and Planetary Science Letters 274, 423-438, 2008.

Cited references

Anderson, R., Fleisher, M., and Lao, Y.: Glacial–interglacial variability in the delivery of dust to the central equatorial Pacific Ocean, Earth and Planetary Science Letters, 242, 406-414, 2006.

Bronk Ramsey, C.: Deposition models for chronological records, Quat. Sci. Rev., 27, 42-60, 2008.

Burckel, P., Waelbroeck, C., Gherardi, J.-M., Pichat, S., Arz, H., Lippold, J., Dokken, T., and Thil, F.: Atlantic Ocean circulation changes preceded millennial tropical South America rainfall events during the last glacial, Geophys. Res. Lett., 42, 411-418, 2015.

Chiang, J. C. H., Biasutti, M., and Battisti, D. S.: Sensitivity of the Atlantic Intertropical Convergence Zone to Last Glacial Maximum boundary conditions, Paleoceanography, 18, doi:10.1029/2003PA000916, 2003.

Croudace, I., and Rothwell, R.: Micro-XRF studies of sediment cores, Dev. Paleoenviron. Res., 17, 2015.

Freeman, E., Skinner, L. C., Waelbroeck, C., and Hodell, D.: Radiocarbon evidence for enhanced respired carbon storage in the Atlantic at the Last Glacial Maximum, Nat Commun, 7, 10.1038/ncomms11998, 2016.

Jaeschke, A., Rühlemann, C., Arz, H., Heil, G., and Lohmann, G.: Coupling of millennial-scale changes in sea surface temperature and precipitation off northeastern Brazil with high-latitude climate shifts during the last glacial period, Paleoceanography, 22, PA4206, doi:4210.1029/2006PA001391, 2007.

Key, R. M., Kozyr, A., Sabine, C. L., Lee, K., Wanninkhof, R., Bullister, J. L., Feely, R. A., Millero, F. J., Mordy, C., and Peng, T. H.: A global ocean carbon climatology: Results from Global Data Analysis Project (GLODAP), Global biogeochemical cycles, 18, 2004.

Lougheed, B. C., Metcalfe, B., Ninnemann, U. S., and Wacker, L.: Moving beyond the age–depth model paradigm in deep-sea palaeoclimate archives: dual radiocarbon and stable isotope analysis on single foraminifera, Climate of the Past, 14, 515-526, 2018.

Reimer, P., Bard, E., Bayliss, A., Beck, J. W., Blackwell, P. G., Bronk Ramsey, C., Buck, C. E., Cheng, H., Edwards, R. L., Friedrich, M., Grootes, P. M., Guilderson, T. P., Haflidason, H., and ... IntCal13 and Marine13 radiocarbon age calibration curves 0–50,000 years cal BP, Radiocarbon, 55, 1869–1887, 2013.

Taylor, S., and McLennan, S.: The continental crust: its composition and evolution, Oxford: Blackwell Press, 312 pp., 1985.

Thornalley, D. J. R., Barker, S., Broecker, W. S., Elderfield, H., and McCave, I. N.: The Deglacial Evolution of North Atlantic Deep Convection, Science, 331, 202-205, 2011.

Vazquez Riveiros, N., Waelbroeck, C., Roche, D. M., Moreira, S., Burckel, P., Dewilde, F., Skinner, L., Böhm, E., Arz, H. W., and Dokken, T.: Northern origin of  western tropical Atlantic deep waters during Heinrich Stadials, submitted.

Waelbroeck, C., Duplessy, J.-C., Michel, E., Labeyrie, L., Paillard, D., and Duprat, J.: The timing of the last deglaciation in North Atlantic climate records, Nature, 412, 724-727, 2001.

Weltje, G. J., and Tjallingii, R.: Calibration of XRF core scanners for quantitative geochemical logging of sediment cores: theory and application, Earth and Planetary Science Letters, 274, 423-438, 2008.

---

## Author Comment (AC2) · 28 Jun 2018

**Detailed response to Referee 2's comments**

We are very grateful for the constructive and helpful comments we received from both reviewers. Accounting for them has been of great help to improve the manuscript.

Waelbroeck et al. have measured Ti/Ca and Pa/Th in a core taken on the margin of northern Brazil, which records rainfall on the nearby continental area and the strength of the Atlantic meridional overturning circulation at intermediate depths. Since the two measurements are done on the same core, they are able to determine the phase relationship between the two variables with minimum uncertainty, which gives them new insights into the response of the ITCZ to changes in the AMOC. They use wavelet analysis to determine phase lags at different frequencies and show that ITCZ movement lags changes in AMOC both at the D/O and HE frequency, but more so at the HE frequency. They attribute this difference to a positive feedback between the strength of the AMOC, seawater temperature and iceberg discharge, which they had first proposed in an earlier publication. The authors pay due attention to the possible impact of bioturbation on the phase relationship between Ti/Ca and Pa/Th, which they convincingly rule out, and to multiple caveats in the interpretation of Pa/Th in term of circulation changes. The paper is clearly written and provides important new findings. I recommend publication after considering the relatively minor comments below (note, however, that I am unable to provide knowledgeable comments on the technicalities of wavelet analysis).

While the authors have clearly established the lag between Ti/Ca and Pa/Th, their ultimate goal is to establish the lag in the response of the ITCZ to changes in AMOC. I think the authors should also discuss the extent to which there might be lags between processes and proxies. For instance, would a change in AMOC translate instantaneously into a change in Pa/Th in their core? I think this is unlikely. Pa/Th recorded in sediments is controlled by the ratio between lateral transport by circulation and vertical transport by scavenging of the Th and Pa produced in the water column. Even if seawater were flowing from the north Atlantic to the Brazilian margin through a pipe (i.e. changes in deep water formation would translate into an instantaneous change in lateral velocity in the pipe), there should still be a lag between sediment Pa/Th and changes in AMOC, depending on the response time of dissolved Th and Pa in the water column overlying the coring site. While the response time of Th is decadal, the full expression on circulation changes on Pa may take several centuries. In addition, the "pipe" is of course an unrealistic cartoon of the AMOC. In reality, I would expect an additional lag between lateral velocity at the coring site and changes in deep water formation, but at this point this is just intuitive and it is well beyond me to guess how long or how short this lag would be. Nonetheless, I think the authors could bring this up and indicate that the lag between Ti/Ca and Pa/Th should be taken as a minimum of the lag of the response of the ITCZ to changes in AMOC. There might also be a lag between Ti/Ca and the change in the seasonal latitudinal range of position of the ITCZ depending on the location of the region supplying lithogenics to the coring site. For instance, if the region is farther south from the southernmost zone of precipitation before the change in AMOC, it may take more time for the ITCZ to reach this region.

We thank Roger François for this important remark. We agree that we should explicitly mention the fact that a change in AMOC does not translate instantaneously into a change in Pa/Th.

To do so, we have added the following few sentences to the paragraph starting at line 25 on p. 13 "[…] our results indicate that rainfall increases in the region adjacent to MD09-3257

occurred several hundred years after the increase in sedimentary Pa/Th at our core site.":

"Furthermore, this lead of sedimentary Pa/Th over ln(Ti/Ca) should be taken as a minimum of the lead of AMOC over ln(Ti/Ca) because a change in AMOC does not translate instantaneously into a change in sedimentary Pa/Th. A delay between a change in AMOC and the resulting change in sedimentary Pa/Th is expected, which depends on the propagation time of the circulation change to the core site and on the response time of dissolved Th and Pa in the water column overlying the core site (i.e. 30-40 for $^{230}$Th, 100-200 y for $^{231}$Pa (François, 2007)). However, increases or decreases in sedimentary Pa/Th should be measurable before the dissolved Th and Pa have fully adjusted to the new circulation regime, especially at sites with high sedimentation rates as at our study site. We thus expect this additional delay to be less than 100 y and much smaller than the computed lead of MD09-3257 sedimentary Pa/Th over ln(Ti/Ca)."

Concerning a possible lag between Ti/Ca and the change in the position of the ITCZ, we have specified in the submitted manuscript that our marine core records can only inform on rainfall changes over the catchment area of the rivers which directly deliver sediment to the study site, that is, over the adjacent continent. Rainfall changes in a region located north of this catchment area may occur before the rainfall changes recorded in our marine cores but we have no means to assess such a delay. We thus prefer not add anything on this subject to text.

While the authors have taken into account how changes in scavenging could obscure the interpretation of Pa/Th in terms of circulation, I think it would also be worth mentioning that interpreting changes in circulation from a single core can also be problematic. While it is correct that higher rate of AMOC should result in a lower sediment Pa/Th when averaged over an entire ocean basin, that may not be correct for any core. Depending on the proximity of the coring location to the site of deep water formation, decreasing the AMOC may actually decrease sediment Pa/Th (e.g. Luo et al., 2010; Fig. 14). I would suggest specifying that we would expect to see an increase in Pa/Th with decreasing rate of AMOC at the coring site of this study because it is sufficiently removed from the site of deep water formation.

Accordingly, I would change the wording on line 4-5 p5: "[*when average over an entire ocean basin*], high (low) flow rates therefore result in high (low) Pa export…"

We are aware that a change in AMOC can produce very different sedimentary Pa/Th signals depending on the location of the cores with respect to that of deep water formation, as demonstrated in Luo et al. (2010).

We have changed the wording on line 4-5 p. 5 as recommended, and modified

"High (low) flow rates therefore result in high (low) Pa export…"

into

"When averaged over an entire ocean basin, high (low) flow rates therefore result in high (low) Pa export…"

Also, we agree that deriving the state of AMOC from a single core location is prone to error. MD09-3257 is however located in an area where the measured Pa/Th vertical profile in core top sediments is consistent with a dominant role of the overturning circulation (Lippold et al., 2011), as explained p. 5, line 11-15 of the submitted manuscript.

Line 26, p7: shorter stadial may have lower increase in Pa/Th because they were too short to allow the full expression of the increase in Pa/Th (limited by the response time of Pa in the

water column).

The Dansgaard-Oeschger (D-O) stadials discussed in the present paper have durations of about 1000 y, which is much longer than the response time of dissolved Pa in the water column (100-200 y for [231]Pa (François, 2007)). Therefore, the response time of dissolved Pa in the water column cannot be the reason for the lower increase in Pa/Th of D-O stadials with respect to Heinrich stadials. Rather, as explained p. 13, lines 27-31 and p. 14 lines 1-2 of the submitted manuscript, we suggest that a positive feedback involving iceberg discharges is only triggered in the case of Heinrich stadials. In contrast, AMOC slowdowns associated with D-O stadials would not trigger such a positive feedback loop and would hence remain limited.

Line 10, p8: (including Pa/Th values susceptible to be partially impacted by large particles flux [*or boundary scavenging resulting from slower AMOC*])

We have added "or boundary scavenging resulting from slower overturning circulation" to the parentheses p. 8, line 10, as recommended.

Line 24, p12: Why not use Th-normalized Ti instead of Ti/Ca to totally eliminate the effect of changes in carbonate dissolution/production?

This is an interesting suggestion. In our manuscript, we relied on the interpretation of the Ti/Ca signal given in former studies of the same area, but our $^{230}Th_{xs,0}$ measurements do indeed give us the opportunity to directly compute Th-normalized Ti flux in order to eliminate the effect of changes in carbonate dissolution and production on the Ti/Ca signal, if any.

We have computed core MD09-3257 Th-normalized Ti signal and compared it with the Ti/Ca and Ti signals in the following figure.

[Figure]

**Figure 1.** Comparison of core MD09-3257 Th-normalized Ti, Ti/Ca and Ti signals.

We see that the Th-normalized Ti signal is indeed very similar to Ti/Ca, thereby confirming

that the Ti/Ca signal is not biased by changes in carbonate dissolution or production.

However, MD09-3257 Th-normalized Ti signal is measured at much lower resolution then MD09-3257 Ti/Ca, and suffers from gaps over the Heinrich stadials. Therefore, the use of MD09-3257 Ti/Ca instead of the Th-normalized Ti signal remains the best option for the present study.

Line 8; p13: Briefly describe what the "independent approach" is.

We have modified the sentence

"This lead is comparable to the relative phase previously estimated between MD09-3257 Pa/Th and Ti/Ca at the onset of HS4 (690 ± 180 y) and HS2 (1420 ± 250 y) respectively, using a completely independent approach (Burckel et al., 2015)."

into

"This lead is comparable to the relative phase previously estimated between MD09-3257 Pa/Th and Ti/Ca at the onset of HS4 (690 ± 180 y) and HS2 (1420 ± 250 y) respectively, based on the identification of the transition in the Pa/Th and Ti/Ca signals at the beginning of these two stadials (Burckel et al., 2015)."

Cited references

François, R.: Chapter Sixteen Paleoflux and Paleocirculation from Sediment 230 Th and 231 Pa/230 Th, Developments in Marine Geology, 1, 681-716, 2007.

Lippold, J., Gherardi, J.-M., and Luo, Y.: Testing the 231Pa/230Th paleocirculation proxy: A data versus 2D model comparison, Geophys. Res. Lett., 38, doi:10.1029/2011GL049282, 2011.

Luo, Y., Francois, R., and Allen, S. E.: Sediment 231Pa/230Th as a recorder of the rate of the Atlantic meridional overturning circulation: insights from a 2-D model, Ocean Science, 6, 381-400, 2010.